# Sensing of nutrients by CPT1C regulates late endosome/lysosome anterograde transport and axon growth

**Marta Palomo-Guerrero[1], Rut Fadó[1], Maria Casas[1], Marta Pérez-Montero[1], Miguel Baena[1], Patrick O Helmer[2], José Luis Domínguez[1], Aina Roig[1], Dolors Serra[3,4], Heiko Hayen[2], Harald Stenmark[5,6], Camilla Raiborg[5,6], Núria Casals[1,4]***

[1]Basic Sciences Department, Faculty of Medicine and Health Sciences, Universitat Internacional de Catalunya, Sant Cugat del Vallès, Spain; [2]Institute of Inorganic and Analytical Chemistry, University of Münster, Münster, Germany; [3]Department of Biochemistry and Physiology, School of Pharmacy, Institut de Biomedicina de la Universitat de Barcelona (IBUB), Universitat de Barcelona, Barcelona, Spain; [4]Centro de Investigación Biomédica en Red de Fisiopatología de la Obesidad y la Nutrición (CIBEROBN), Instituto de Salud Carlos III, Madrid, Spain; [5]Centre for Cancer Cell Reprogramming, Faculty of Medicine, University of Oslo, Oslo, Norway; [6]Department of Molecular Cell Biology, Institute for Cancer Research, Oslo University Hospital, The Norwegian Radium Hospital, Oslo, Norway

**Abstract** Anterograde transport of late endosomes or lysosomes (LE/Lys) is crucial for proper axon growth. However, the role of energetic nutrients has been poorly explored. Malonyl-CoA is a precursor of fatty acids, and its intracellular levels highly fluctuate depending on glucose availability or the energy sensor AMP-activated protein kinase (AMPK). We demonstrate in HeLa cells that carnitine palmitoyltransferase 1C (CPT1C) senses malonyl-CoA and enhances LE/Lys anterograde transport by interacting with the endoplasmic reticulum protein protrudin and facilitating the transfer of Kinesin-1 from protrudin to LE/Lys. In cultured mouse cortical neurons, glucose deprivation, pharmacological activation of AMPK or inhibition of malonyl-CoA synthesis decreases LE/Lys abundance at the axon terminal, and shortens axon length in a CPT1C-dependent manner. These results identify CPT1C as a new regulator of anterograde LE/Lys transport in response to malonyl-CoA changes, and give insight into how axon growth is controlled by nutrients.

**\*For correspondence:** ncasals@uic.es

**Competing interests:** The authors declare that no competing interests exist.

## Introduction

During the development of the nervous system, axon growth and guidance is known to be tightly regulated by soluble guidance cues, such as semaphorines, ephrins, etc., but also by the proper regulation of intracellular transport of molecules and organelles (*Tymanskyj et al., 2018*). In addition to the recycling endosomes, late endosomes/lysosomes (LE/Lys) can also be recycled to the cell surface in sites of protrusion formation to provide extra membrane for neurite outgrowth (*Raiborg et al., 2016*). Recently, the role of protrudin in protrusion growth in neurons and other cells has been described (*Raiborg et al., 2015*; *Shirane and Nakayama, 2006*). Protrudin is an ER integral membrane protein that interacts with the GTPase Rab7 and phosphatidylinositol-3-phosphate (PI3P) at the LE/Ly membrane and transfers kinesin-1 to the Rab7-associated FYCO1 to mediate plus-end microtubule based transport of LE/Lys. Interestingly, LE/Lys contact the ER repeatedly during their anterograde movement suggesting the existence of some kind of check-point regulation at ER-LE

contact sites (*Wijdeven et al., 2015*). Indeed, amino acids have recently been described to play a regulatory role. In sufficient nutrient conditions, the presence of amino acids facilitates protrudin-mediated translocation of LE/Lys to the cell periphery for mTORC1 activation, and consequently, for protein synthesis and cell growth (*Hong et al., 2017*). By contrast, when amino acids are not available, LE/Lys cluster in the perinuclear area in order to fuse with autophagosomes and replenish the cell with amino acids derived from the digestion of damaged organelles (*Cabukusta and Neefjes, 2018*; *Korolchuk et al., 2011*). On the other hand, cholesterol has been demonstrated to be a regulator of the minus-end transport (*Rocha et al., 2009*; *Wijdeven et al., 2015*). High endosomal cholesterol levels, promote minus-end transport of LE/Lys, whereas under low-endosomal cholesterol conditions, the cholesterol sensing protein ORP1L at the LE/Ly membrane changes its conformation and favours the establishment of extensive ER-LE/Ly contact sites and the release of the minus-end-directed motor complex dynactin-dynein, which stops the retrograde transport. Collectively, data indicate that amino acids and cholesterol are distinct regulators of LE/Lys position and motility.

Even though axon growth is a highly energy-consuming process, no regulatory role has been assigned to the energetic nutrients par excellence, glucose and fatty acids, on LE/Lys transport. Moreover, metabolic stresses that regulate axon growth are poorly understood. It is known that the master energy sensor 5′-AMP activated protein kinase (AMPK) regulates axon growth in developing neurons. AMPK is activated to maintain cellular energy homeostasis by switching off ATP-consuming processes. Specifically during neuron development, activated AMPK phosphorylates the motor protein KIF5 preventing PI3K transport to the axon tip, necessary for axon growth (*Amato et al., 2011*). However, the roles of other well know down-stream effectors of the AMPK pathway, such as acetyl-CoA carboxylase (ACC), have not been studied. ACC synthesizes malonyl-CoA, the precursor of fatty acids, when glucose is abundant in the culture media. By contrast, in energy stressful conditions, AMPK phosphorylates and inhibits ACC (*Hardie and Pan, 2002*). In consequence, malonyl-CoA levels highly fluctuate depending on the energy status of the cell, being high after feeding and low upon fasting (*Tokutake et al., 2012*; *Tokutake et al., 2010*). Since malonyl-CoA is in the crossroad of fatty acids and glucose metabolism, it emerges as an interesting metabolite to be considered as a putative regulator of LE/Lys transport.

On the other hand, malonyl-CoA is the physiological regulator of carnitine palmitoyltransferase 1 (CPT1), the rate-limiting enzyme of mitochondrial fatty acid oxidation (*Saggerson, 2008*). Interestingly, the neuron-specific CPT1 isoform, CPT1C, is thought to be a pseudoenzyme because it is localized to the ER, and has minimal catalytic activity compared to the mitochondrial isoforms CPT1A and CPT1B (*Sierra et al., 2008*). Since CPT1C maintains the ability to bind malonyl-CoA, we and other groups have postulated that CPT1C could behave as a sensor of malonyl-CoA in neurons (*Price et al., 2002*; *Wolfgang and Lane, 2011*; *Casals et al., 2016*). The development of Cpt1c knock-out (KO) mice has shed light on the physiological role of CPT1C in neurons. *Cpt1c* KO mice show motor function deficits, such as ataxia, dyscoordination, and muscle weakness (*Carrasco et al., 2013*), in addition to learning deficits (*Carrasco et al., 2012*) and impaired hypothalamic control of body energy homeostasis (*Casals et al., 2016*; *Pozo et al., 2017*; *Rodríguez-Rodríguez et al., 2019*). Interestingly, the unique two CPT1C mutations described in humans to date have been associated with hereditary spastic paraplegia (HSP) (*Hong et al., 2019*; *Rinaldi et al., 2015*). HSPs are a group of inherited neurological disorders characterized by slowly progressive weakness and spasticity of the muscles of the legs, caused by axonopathy of corticospinal motor neurons (*Blackstone et al., 2011*). Of note, Impairment in organelle transport along the axon is a common trait in the development of the disease (*Boutry et al., 2019*).

In the present study, we explore the role of CPT1C as a sensor of malonyl-CoA in the regulation of axon growth in response to nutritional changes. Our results show that CPT1C is necessary for proper axon growth and identify the malonyl-CoA/CPT1 axis as a new regulator of LE/Lys antero-grade transport. Under normal nutrient conditions, CPT1C promotes the anterograde transport of LE/Lys by enhancing protrudin-mediated transfer of the motor protein kinesin-1 to LE/Lys; while under energy stress, which leads to a decrease in malonyl-CoA levels, CPT1C stops this enhancement and the plus-end motion is arrested. The regulation of LE/Lys positioning in response to intracellular malonyl-CoA is crucial for proper regulation of axon growth in cortical neurons and can give new clues for the understanding of HSP.

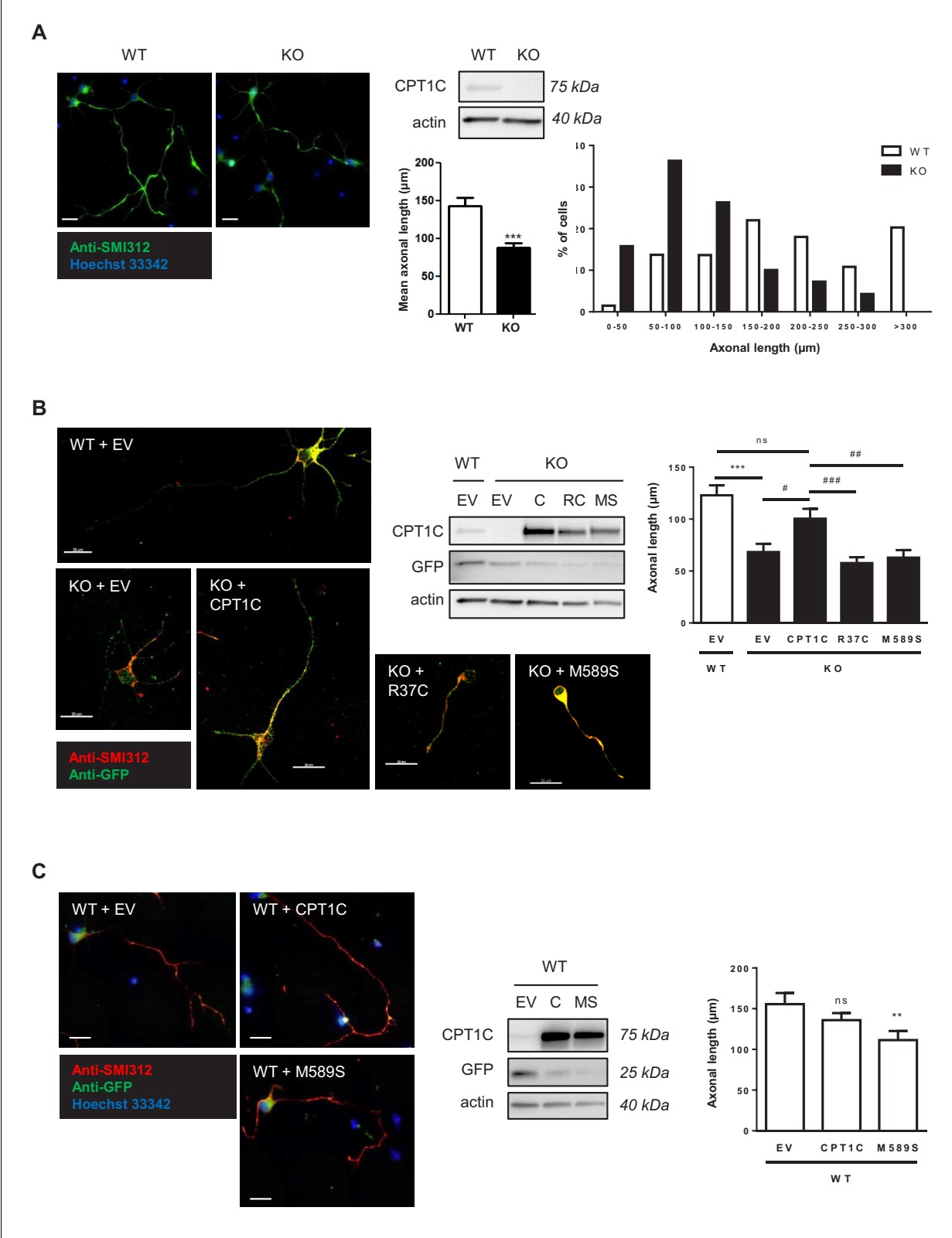

**Figure 1.** CPT1C is necessary for proper axon growth. (**A**) Primary cortical neurons derived from WT and *Cpt1c* KO E16 mouse embryos were cultured and fixed at 4DIV. Then, axons were labeled with a specific marker (SMI-312; in green) and nuclei were detected with Hoechst staining (blue). CPT1C absence in KO cultures was corroborated by western blot. Axonal length was analyzed from three independent experiments performed in biological triplicates. Right graph shows the percentage of cells with axons of a certain length (intervals of 50 μm), while in left graph the mean ± SEM of all axons

*Figure 1 continued on next page*

*Figure 1 continued*

is shown (n = 100 cells per genotype; Student's t test; ***p<0.001). (**B**) *Cpt1c* KO neurons were infected at 1DIV with lentiviral vectors that codified for mouse CPT1C or the mutated forms M589S (MS) or R37C (RC). At 4DIV, cells were fixed and axon was identified as described above. GFP was used to detect infected cells. Immunoblotting was performed to confirm CPT1C and M589S expression in infected KO neurons. Graph shows the mean axonal length ± SEM of 2 independent experiments performed in biological duplicates (n = 50 cells per condition; One-way ANOVA followed by Bonferroni's comparison test; ***p<0.001 *versus* WT + EV and #p<0.05, ##p<0.01 and ###p<0.001 *versus* KO + CPT1C). (**C**) Effect of M589S overexpression in WT cells. Graph shows the mean axonal length ± SEM of 2 independent experiments performed in biological duplicates (n = 50 cells per condition; One-way ANOVA followed by Bonferroni's comparison test; **p<0.01 *versus* WT + EV). Scale bar, 20 μm.

The online version of this article includes the following figure supplement(s) for figure 1:

**Figure supplement 1.** Dendritic arborization of *Cpt1c* KO cortical neurons.

## Results

### CPT1C is necessary for proper axon growth

Since CPT1C has been associated with HSP, we decided to study whether CPT1C is necessary for proper axon growth. Cultured cortical neurons derived from wild type (WT) and *Cpt1c*KO mice were analyzed at 4 days of in vitro division (DIV) and we observed that *Cpt1c*KO neurons had shorter axons (about a 40% length reduction) compared to WT neurons (*Figure 1A*). Then, rescue experiments were performed with WT CPT1C and two mutated proteins: CPT1C$^{R37C}$, the missense mutation associated with HSP (*Rinaldi et al., 2015*); and CPT1C$^{M589S}$, a mutation that maintains CPT1 functionality but prevents the binding of malonyl-CoA (*Morillas et al., 2003*; *Rodríguez-Rodríguez et al., 2019*). Axon length was rescued by CPT1C re-expression but not by CPT1C$^{R37C}$ or CPT1C$^{M589S}$ (*Figure 1B*). These results suggested that the binding of malonyl-CoA to CPT1C is required for proper axon growth. Accordingly, CPT1C$^{M589S}$ overexpression in WT cultures reduced axon length (*Figure 1C*). Of note, CPT1C overexpression in WT cultures was not enough to increase axon length, indicating that CPT1C is necessary but not sufficient for promoting axon growth.

We also analysed dendritic arborization at 4DIV and found that it was diminished in *Cpt1c*KO neurons compared to WT cells (*Figure 1—figure supplement 1A–B*).). Moreover, the percentage of polarized cells (cells with a defined axon between the neurites) was reduced, and the percentage of cells without any neurite was increased in *Cpt1c* KO cultures (*Figure 1—figure supplement 1C*). CPT1C overexpression in WT neurons was not enough to increase dendritic arborization (*Figure 1—figure supplement 1D*), as observed with axon growth. In summary, our results indicate that CPT1C is necessary for proper axon growth, cell polarization and dendritic arborization.

### Axon growth is arrested by malonyl-CoA synthesis inhibition

We next asked whether a decrease in malonyl-CoA levels was able to arrest axon growth in WT and *Cpt1c* KO neurons. To address this, three different short treatments, known to reduce malonyl-CoA synthesis, were performed in primary cortical neurons at 3DIV: 1) cell media were depleted of glucose for 2 hr; 2) cells were treated with 5-aminoimidazole-4-carboxamide ribonucleotide (AICAR, an activator of AMPK) for 1.5 hr; or 3) cells were treated with 5-(tetradecyloxy)−2-furoic acid (TOFA, an inhibitor of ACC) (*Halvorson and McCune, 1984*) for 1.5 hr. Axon length was measured 24 hr later. We checked that glucose depletion or AICAR treatment increased ACC phosphorylation, as indicative of malonyl-CoA synthesis inhibition (*Figure 2A*). All the treatments reduced axon length in WT cortical neurons suggesting a key role for malonyl-CoA in the regulation of axon growth (*Figure 2B*). However, none of the treatments were able to further reduce axon length in *Cpt1c* KO neurons, pointing to CPT1C as a mediator of the malonyl-CoA effects.

### CPT1C interacts with protrudin

We next aimed to unravel the mechanism by which CPT1C was regulating axon growth. As CPT1C had been identified as one of the putative interactors of protrudin in a proteomic screening (*Hashimoto et al., 2014*), and protrudin was described to promote neurite formation by enhancing the anterograde transport of LE/Lys (*Raiborg et al., 2015*; *Shirane and Nakayama, 2006*), we decided to further study the potential interaction of these two proteins by co-immunoprecipitation (co-IP). HeLa cells were co-transfected with Myc-CPT1C and mCitrine-protrudin or mCitrine-ER-5

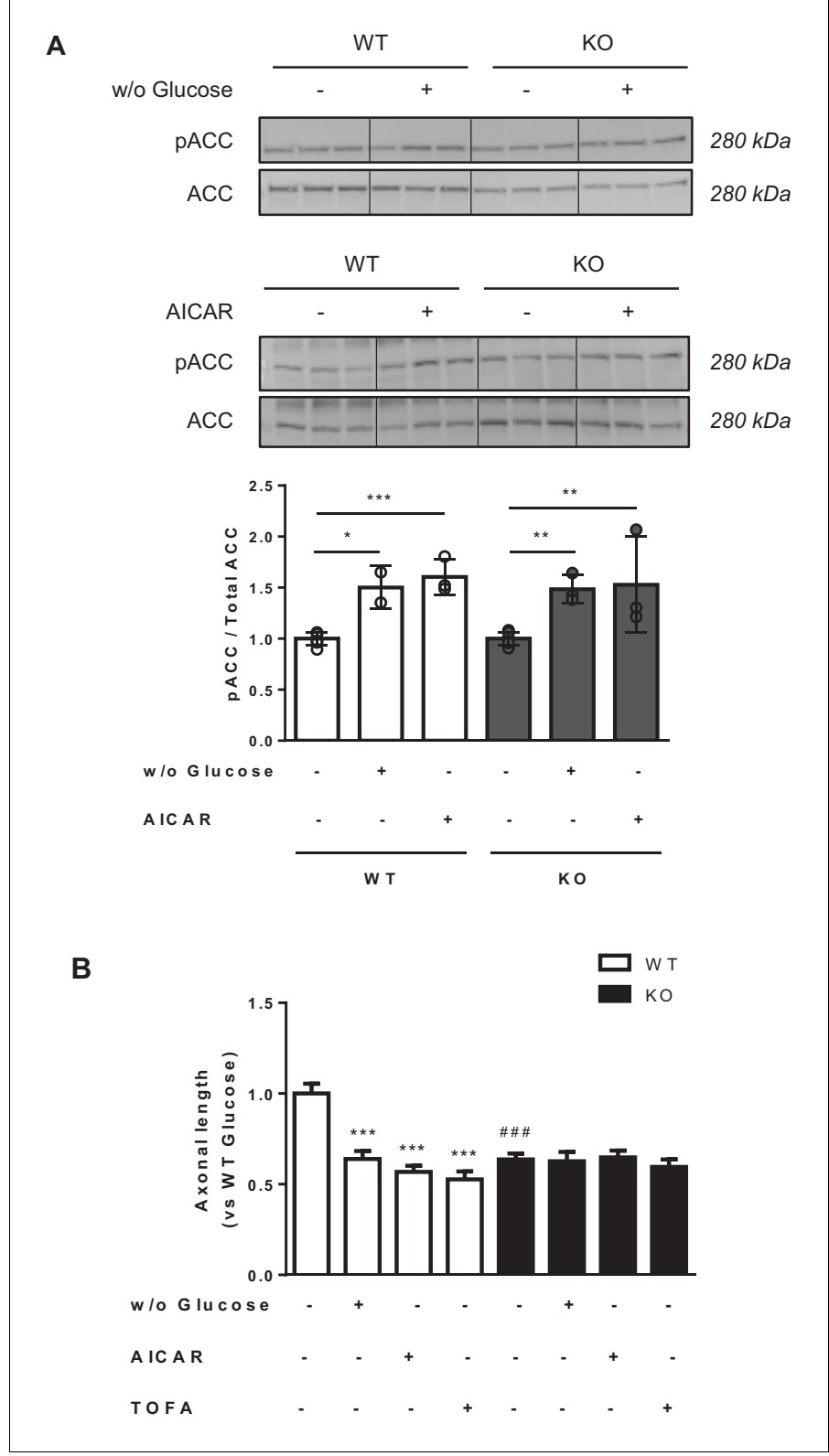

**Figure 2.** Metabolic stress decreases axon growth through CPT1C. 3DIV primary cortical neurons derived from WT and KO CPT1C embryos were incubated with DMEM medium with (25 mM) or without glucose (2 hr), treated with AICAR (1.5 hr at 2.5 mM), TOFA (1.5 hr at 20 μg/mL) or vehicle (DMSO 1:500). At the end of the treatment, the medium was replaced by Neurobasal + B27 conditioned medium. 24 hr later (at 4DIV), cells were fixed and

*Figure 2 continued on next page*

*Figure 2 continued*

processed by immunostaining with anti-SMI-312. (**A**) ACC phosphorylation (inhibition) by glucose depletion or AICAR treatment in WT and *Cpt1c* KO neurons was corroborated by Western blot. For quantification, the ratio between pACC and total ACC levels was used. Results are the mean ± SD of biological triplicates (n = 3 samples per condition; Two-way ANOVA followed by Bonferroni's multiple comparison test; *p<0.05, **p<0.01 and ***p<0.001 *versus* control treatment within the same genotype). (**B**) Mean axon length quantification. The neurofilament marker SMI-312R was used as axon marker. Results are shown as the mean ± SEM of 3 independent experiment performed in biological duplicates (n = 35–60 cells per condition; Two-way ANOVA followed by Bonferroni's multiple comparison test; ***p<0.001 *versus* control treatment and ###p<0.001 *versus* WT within the same treatment).

The online version of this article includes the following source data for figure 2:

**Source data 1.** Source data for pACC/ACC analysis (panel A of the figure).

(contains a KDEL sequence that drives the fluorescent protein to the ER, used as a negative control of the co-IP). With the GFP-Trap assay, protrudin was able to co-immunoprecipitate CPT1C (*Figure 3A*), confirming the interaction between these two proteins. We next performed FRET analysis in HEK293 cells transiently transfected with CPT1C-mTurquoise2 and protrudin-SYFP2. Calreticulin, tubulin and mTurquoise-ER were used as negative interactors of CPT1C, while the pairs atlastin-1/protrudin, and atlastin-1/CPT1C were used as positive controls (*Chang et al., 2013*; *Hashimoto et al., 2014*; *Rinaldi et al., 2015*). Results shown in *Figure 3B–C* confirm that CPT1C is able to interact with protrudin. In order to know whether malonyl-CoA could be regulating this interaction, we performed the same analysis in HEK293 cells submitted to glucose depleted conditions, which activates the AMPK/ACC pathway (*Figure 3D*), or in HeLa cells treated with TOFA. The interaction between CPT1C and protrudin was not affected by glucose availability (*Figure 3E*) or the inhibition of malonyl-CoA synthesis (*Figure 3F*).

## CPT1C regulates anterograde transport of LE/Lys depending on malonyl-CoA levels

We next analysed whether CPT1C could be involved in LE/Lys distribution, as previously described for protrudin (*Raiborg et al., 2015*). To that end, we transfected HeLa cells with CPT1C-mTurquoise2, CPT1C$^{M589S}$-mTurquoise2, mTurquoise-ER (negative control), or mCitrine-protrudin (positive control), and LE/Lys were stained for Lamp-1 and analyzed by fluorescence microscopy. The percentage of cells with Lamp1-positive vesicles in any of their protrusion tips was quantified. *Figure 4* demonstrates that CPT1C overexpression favoured a more peripheral location of LE/Lys compared to control cells, a phenotype similar to protrudin overexpression. Specifically, the percentage of cells with Lamp1 in the protrusion tips increased from 28% in control cells (mTurquoise-ER-transfected cells) to 76% in CPT1C-overexpressing cells. In contrast, the expression of CPT1C$^{M589S}$ did not trigger any change in LE/Lys distribution compared to control cells. Interestingly, the treatment of cells with TOFA for 1 hr, which drastically reduced the level of malonyl-CoA (*Figure 4A*), was enough to avoid the peripheral distribution of LE/Lys in CPT1C-overexpresssing cells, but not in cells that overexpressed protrudin (*Figure 4B–C*). Moreover, cell circularity was increased by TOFA treatment in CPT1C-overexpressing cells (*Figure 4D*), which is indicative of decreased protrusion formation.

Live tracking of individual LE/Lys in HeLa cells was performed next. For this purpose, we overexpressed mCherry-FYCO1, which binds coincidently to Rab7 and PI3P in LE/Lys and participates actively in the anterograde transport of the vesicle. mCherry-FYCO1-positive vesicles were live tracked and velocity, distance and directionality were measured (*Figure 5* and *Figure 5—videos 1*, *2*, *3*, *4*, *5* and *6*). These analyses showed that CPT1C overexpression favored anterograde transport of LE/Lys, but had no effect on retrograde transport. Specifically, CPT1C overexpression increased the anterograde moving distance (*Figure 5A–B*), while not modifying the velocity (*Figure 5C*). TOFA treatment reduced the moving distance, velocity and the directionality only in CPT1C-expressing cells (*Figure 5A–E*). TOFA also showed a tendency to reduce the anterograde velocity in control cells and the retrograde velocity in both control and CPT1C-overexpressing cells, which could be suggestive of other TOFA effects independent of CPT1C. As expected, CPT1C$^{M589S}$ expression

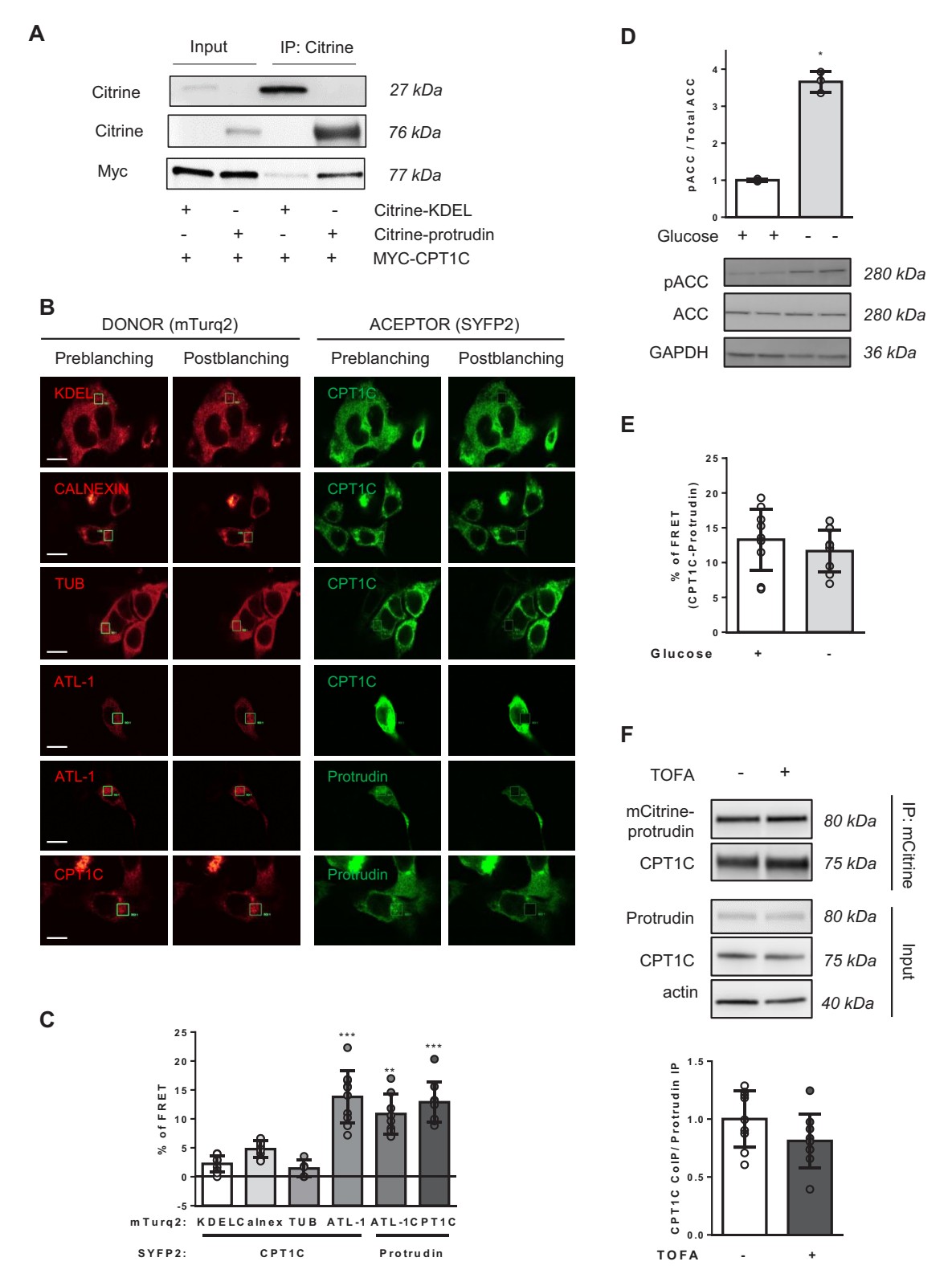

**Figure 3.** CPT1C-protrudin interaction. (**A**) CPT1C-protrudin binding was analyzed in HeLa cells by co-immunoprecipitation (co-IP). HeLa cells were co-transfected with mCitrine-protrudin or mCitrine-ER-5 (Citrine-KDEL) and MYC-CPT1C. 36 hr later, citrine-containing proteins were immunoprecipitated with the GFP-Trap assay, and the indicated proteins were detected by immunoblot in whole lysate (input) and immunoprecipitated samples (IP). The co-IP was performed in biological triplicates. A representative image of the experiment is shown. (**B–C**) Binding evaluation by FRET assay in HEK293

*Figure 3 continued on next page*

*Figure 3 continued*

cells. Percentage of FRET was measured by the increase of donor intensity after photobleaching. mTurquoise-ER (KDEL), mTurquoise-calnexin and Tubulin-mTurquoise2 (TUB) with CPT1C-SYFP2 were used as negative interactions, while Atlastin1-mTurquoise2 (ATL1) with protrudin-SYFP2 and atlastin-1-mTurquoise2 with CPT1C-SYFP2, as positive controls. Representative images of transfected cells with proteins fused to mTurquoise2 (donor; red) or SYFP2 (acceptor; green) are shown in B. Scale bar, 20 µm. Values are shown in C as mean ± SD of 2 independent experiments performed in biological duplicates (n = 5–11 cells per condition were analyzed; One-way ANOVA followed by Bonferroni's multiple comparison test; **p<0.001 and ***p<0.001 vs KDEL control). (D) ACC phosphorylation after 4 hr of glucose deprivation in HEK293 cells was confirmed by Western blot. Graph shows the mean ± SD of biological triplicates (n = 3 samples per condition; Mann-Whitney U test; *p=0.05). (E–F) CPT1C-protrudin interactions under metabolic stress conditions. In E, HEK293 cells were incubated in complete medium (25 mM of glucose) or with a medium without glucose (4 hr) and CPT1C-protrudin interaction was analyzed by FRET assay. Results are given as mean ± SD of 2 independent experiments performed in biological duplicates (n = 9–10 cells per condition; Student's t test; p=0.4315). In F, CPT1C stably expressing HeLa cells, were transfected with mCitrine-protrudin for 24 hr and treated with TOFA (1 hr at 20 µg/mL). Immunoprecipitation was used to analyze CPT1C-protrudin binding. Results are given as mean ± SD of 2 independent experiments performed in biological quadruplicates or quintuplicates (n = 9 samples per condition; Student's t test; p=0.1142).
The online version of this article includes the following source data for figure 3:

**Source data 1.** Percentage of FRET between different donor and acceptor proteins (panel C of the figure).
**Source data 2.** Source data for pACC/ACC analysis (panel D of the figure).
**Source data 3.** Source data for co-immunoprecipitation analysis (panel E of the figure).
**Source data 4.** Source data for co-immunoprecipitation analysis (panel F of the figure).

reduced the anterograde moving distance and velocity of mCherry-FYCO1 vesicles to the same extent as TOFA treatment.

These results indicate that CPT1C senses malonyl-CoA and regulates anterograde transport of LE/Lys. When malonyl-CoA levels are high, the anterograde transport is promoted, but when they decrease, the plus-end motility of FYCO1-positive vesicles is slowed down or inhibited.

## Protrudin-meditated transfer of kinesin-1 to LE/Lys is enhanced by CPT1C when malonyl-CoA is available

Next, we aimed to study whether CPT1C was in close contact with LE/Lys. For that purpose, we overexpressed CPT1C-mTurquoise2 and mCitrine-protrudin in HeLa cells and performed immunostaining using antibodies against Lamp1 (*Figure 6*). Confocal microscopy confirmed the localization of CPT1C in the ER, using VAPA as an ER marker. Moreover, CPT1C was found in ER-lysosome contact sites partly colocalizing with protrudin. CPT1C was not enriched in contact sites compared to protrudin, which was highly enriched there. These findings are consistent with the notion that CPT1C does not form the contact sites, but serves as a regulator of their function.

Then, we analysed whether the CPT1C-mediated regulation of LE/Lys transport was dependent on protrudin. Protrudin silencing in CPT1C-expressing cells highly reduced the percentage of cells with Lamp1 in the periphery (*Figure 7A* and *Figure 7—figure supplement 1A*). These results indicate that CPT1C effects are mediated, at least partially, by protrudin. Interestingly, the reduction driven by protrudin silencing was similar to the reduction observed by TOFA treatment (*Figure 7B*), and TOFA did not show any further effect in protrudin-silenced cells. These data suggest that malonyl-CoA/CPT1C axis and protrudin share the same signalling pathway.

As protrudin establishes ER-LE/Lys contacts and recruits kinesin-1 we decided to analyse whether the number of contacts, or the intensity of kinesin-1 recruited to protrudin was regulated by CPT1C or malonyl-CoA. The results showed that the fraction of Lamp1 or Rab7 overlapping with protrudin was neither changed by CPT1C-overexpression nor by TOFA treatment (*Figure 7C* and *Figure 7—figure supplement 1B*, respectively), indicating that protrudin-mediated ER-LE/Lys contacts are not regulated by the malonyl-CoA/CPT1C axis. Next, we analysed kinesin-1 (KIF5B) recruitment to protrudin by co-immunoprecipitation. Interestingly, CPT1C overexpression reduced protrudin-association with kinesin-1, but CPT1C^M589S did not (*Figure 7D*). These results suggest that the malonyl-CoA bound conformation of CPT1C favours kinesin-1 transfer from ER-bound protrudin to LE/Lys.

To verify this notion, we analyzed the intensity of kinesin-1 in Lamp1-positive vesicles and found that it was enhanced by CPT1C overexpression, while the concomitant treatment with TOFA blocked that increase (*Figure 7E*). Moreover, in cells overexpressing FYCO1, CPT1C increased kinesin-1 recruitment to FYCO1-positive vesicles, but not when the FYCO1 mutant (Δ735–773) that lacks the kinesin-1-binding domain was used (*Figure 7—figure supplement 1C*).

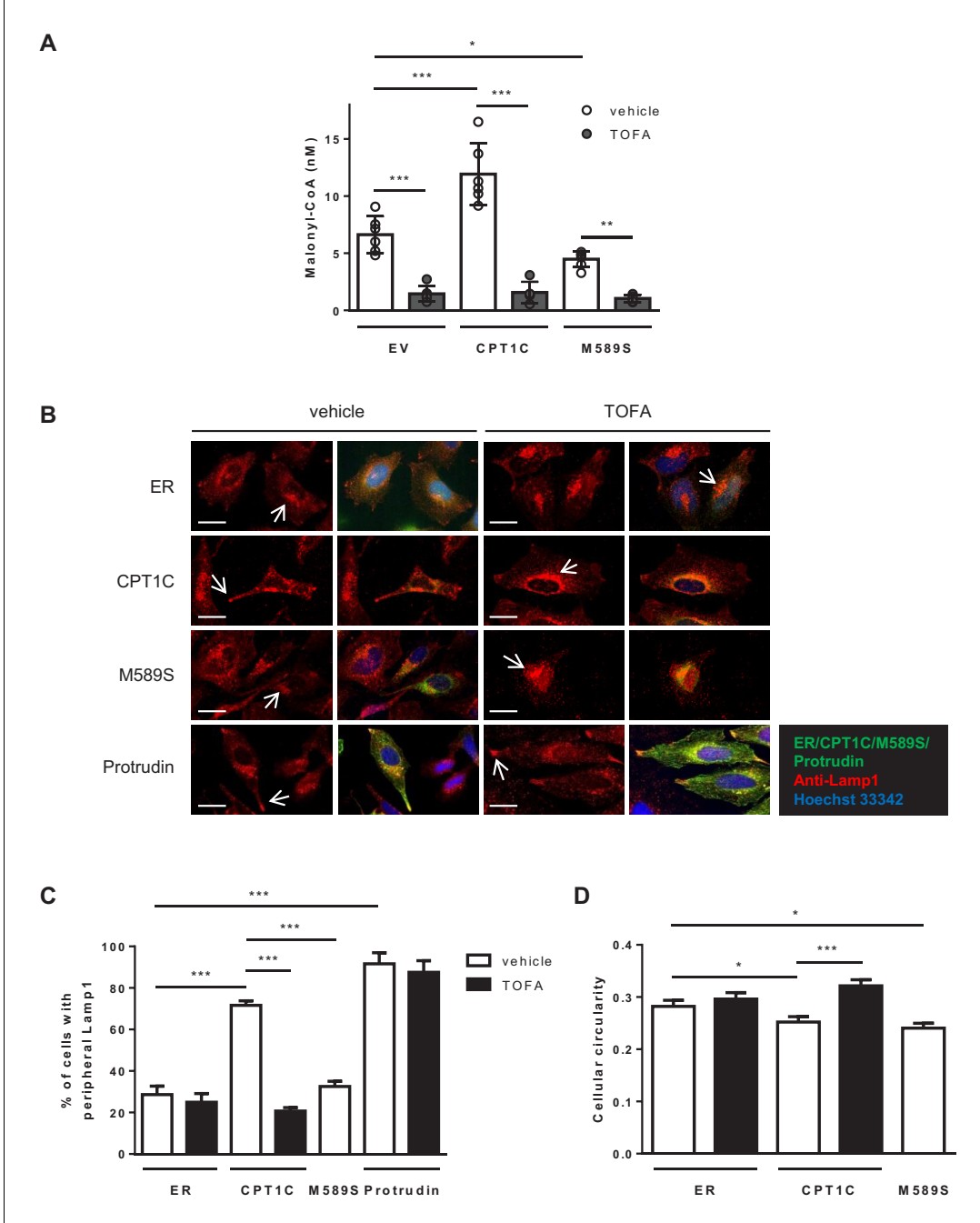

**Figure 4.** CPT1C overexpression promotes the peripheral localization of Lamp1-positive vesicles depending on malonyl-CoA levels. (**A**) Intracellular malonyl-CoA levels decrease after TOFA treatment. HeLa cells stably expressing empty vector, CPT1C or CPT1C$^{M589S}$ were treated with TOFA (1 hr at 20 µg/mL) or the vehicle (DMSO; 1:500). Then, cells were collected and processed as explained in Materials and methods section for malonyl-CoA levels determination. Raw data are the mean of technical duplicates. Results are the mean ± SD of 2 independent experiments performed in biological triplicates (n = 4–6 total samples; Two-way ANOVA followed by Bonferroni's multiple comparison test; *p<0.05, **p<0.01 and ***p<0.001). (**B**) Representative images of HeLa cells transfected with CPT1C-mTuquoise2, CPT1C$^{M589S}$-mTurquoise2 or mTurquoise-ER (shown in green) and immunostained with anti-Lamp1 (in red). Scale bar, 20 µm. White arrows show the accumulation of Lamp1-positive vesicles. (**C**) Percentages of cells with peripheral localization of Lamp1. (**D**) Quantification of cellular circularity. Values in C and D are shown as mean ± SEM of 5 independent experiments performed in biological duplicates (n = 100 cells per condition were analyzed; One-way ANOVA, followed by Bonferroni's comparison test; *p<0.05 and ***p<0.001).

The online version of this article includes the following source data for figure 4:

**Source data 1.** Source data for malonyl-CoA analysis in cells (panel A of the figure).

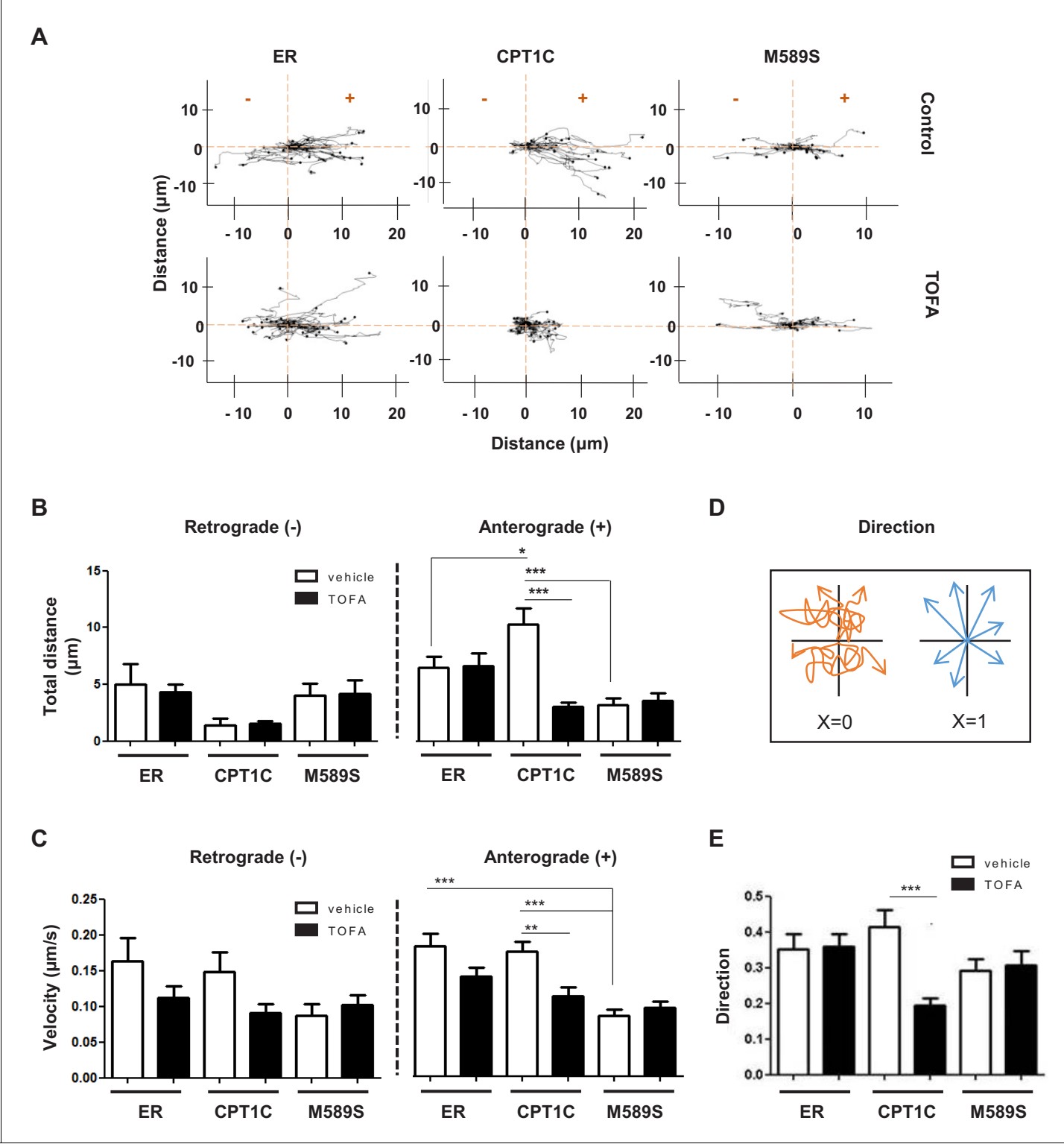

**Figure 5.** CPT1C regulates the anterograde transport of LEs depending on malonyl-CoA levels. HeLa cells were co-transfected with mTurquoise-ER, CPT1C-mTurquoise2 or CPT1C[M589S]-mTurquoise2, and mCherry-FYCO1. After 48 hr, cells were treated with TOFA (1 hr at 20 µg/mL) or vehicle (control, DMSO 1:500) for 1 hr. Trajectories of mCherry-FYCO1 vesicles were analyzed. Representative videos are shown in *Figure 5—videos 1–6*. (**A**) Movement pattern of individual mCherry-FYCO1 vesicles in two dimensions. (**B**) Total distance traveled in anterograde and retrograde directions. (**C**) Quantification of vesicle velocity. (**D**) Scheme for the direction quantification. The random movement of the vesicles was considered value 0, and the directional movement value 1. (**E**) Analysis of vesicle directionality. Results are shown as mean ± SEM of 3 independent experiments performed in
*Figure 5 continued on next page*

*Figure 5 continued*

biological duplicates (n = 30–40 vesicles per condition were analyzed; Two-way ANOVA followed by Bonferroni's comparison test; *p<0.05, **p<0.01 and ***p<0.001).

The online version of this article includes the following video(s) for figure 5:

**Figure 5—video 1.** Representative video of FYCO-positive vesicles in a control HeLa cell.
https://elifesciences.org/articles/51063#fig5video1

**Figure 5—video 2.** Representative video of FYCO-positive vesicles in a control HeLa cell treated with TOFA.
https://elifesciences.org/articles/51063#fig5video2

**Figure 5—video 3.** Representative video of FYCO-positive vesicles in a HeLa cell over-expressing CPT1C.
https://elifesciences.org/articles/51063#fig5video3

**Figure 5—video 4.** Representative video of FYCO-positive vesicles in a HeLa cell over-expressing CPT1C, after TOFA treatment.
https://elifesciences.org/articles/51063#fig5video4

**Figure 5—video 5.** Representative video of FYCO-positive vesicles in a HeLa cell over-expressing CPT1C$^{M589S}$.
https://elifesciences.org/articles/51063#fig5video5

**Figure 5—video 6.** Representative video of FYCO-positive vesicles in a HeLa cell over-expressing CPT1C$^{M589S}$, after TOFA treatment.
https://elifesciences.org/articles/51063#fig5video6

In summary, the overall results indicate that CPT1C enhances protrudin-mediated transfer of kinesin-1 to FYCO1 at LE/Lys only in the presence of malonyl-CoA.

## The malonyl-CoA/CPT1C axis regulates LE/lys abundance at the axon tip

Next, we studied whether CPT1C and malonyl-CoA exerted a regulatory role on LE/Lys transport in cortical neurons. Lamp1 intensity at the growth cone of individual cultured cortical neurons at 4 DIV was analysed by confocal microscopy. We compared WT and *Cpt1c*KO cells under glucose depletion, AMPK activation by AICAR, or TOFA treatment. The mean intensity, volume and total intensity of the most apical Lamp1 positive dot accumulation was quantified (*Figure 8A–B* and *Figure 8—figure supplement 1A*). Since the morphology of the growth cone and the volume of the most apical Lamp1 dot accumulation showed large differences between cells, we decided to use the mean intensity as a good measure to normalize the other values. Results showed that the mean intensity of Lamp1 in *Cpt1c*KO neurons was reduced compared to WT cells (*Figure 8B*). Moreover, CPT1C re-expression in KO neurons restored the mean intensity of Lamp1 at the axon tip (*Figure 8—figure supplement 1B*). Glucose deprivation, AICAR or TOFA treatment diminished Lamp1 mean intensity at growth cones in WT neurons, but none of the treatments was able to further decrease it in *Cpt1c* KO cells, in accordance with the measurements of axon length (*Figure 2*). We also performed a linear Lamp1 fluorescence profile throughout 50 µm from the axon tip towards the soma. In non-treated WT neurons, Lamp1 intensity at the tip was highly increased compared to the rest of the axon (*Figure 8C*). By contrast, in treated WT neurons or *Cpt1c* KO neurons Lamp1 intensity was very similar throughout the axon. Only WT neurons treated with TOFA showed an increase in Lamp1 intensity closer to the soma. Collectively, these data indicate that energy stress that decreases malonyl-CoA levels causes a CPT1C-mediated reduction in the number of LE/Lys that reach the axon tip, resulting in axon growth arrest.

To confirm the critical role of malonyl-CoA in LE/Lys anterograde transport and axon growth, we next analyzed whether the addition of malonyl-CoA to neuron cultures concomitantly to glucose depletion, AICAR or TOFA treatment, reversed the effects of these treatments. We first checked that malonyl-CoA was able to penetrate the cells and increase its intracellular levels (*Figure 9A*), as previously described (*Knobloch et al., 2017*). *Figure 9B* shows that exogenous malonyl-CoA was enough to partially reverse Lamp1 intensity decrease at the axon tip, caused by glucose depletion or TOFA treatment (a tendency was observed in AICAR treated cells). Moreover, axon growth was partially restored in all conditions (*Figure 9C*). However, malonyl-CoA did not cause any effect in CPT1C deficient cells. We conclude that the malonyl-CoA/CPT1C axis is an important regulator of LE/Lys anterograde transport and axon growth in response to energy stress.

## Discussion

Collectively our data demonstrate that CPT1C is a new regulator of LE/Lys anterograde transport in neurons. Under sufficient nutrient supply, CPT1C will sense intracellular malonyl-CoA levels and promote the plus-end transport of LE/Lys through the enhancement of protrudin function. By contrast, under energy stresses, such as glucose depletion or AMPK activation, malonyl-CoA synthesis will be inhibited, and the malonyl-CoA-unbound CPT1C will prevent the protrudin-mediated transfer of kinesin-1 to LE/Lys. As a consequence, axon growth in neurons will be temporarily arrested (see *Figure 10* for a scheme).

Our results also show that CPT1C is needed for proper dendrite growth, since CPT1C KO neurons showed poor arborisation. However, we do not think this effect is mediated by protrudin, since it has been demonstrated that protrudin promotes the growth of axon but not of dendrites (*Hashimoto et al., 2014*).Therefore, we believe that CPT1C could play a role in dendrite arborization through a mechanism independent of protrudin.

LE/Lys positioning regulates many diverse cellular responses (*Korolchuk et al., 2011*; *Neefjes et al., 2017*). Results shown here complete the regulatory role previously described for amino acids on LE/Lys positioning. The presence of nutrients in the media will increase intracellular malonyl-CoA in addition to amino acides, and in consequence favours mobile LE/Lys to move towards the periphery for mTORC activation, local cell growth and membrane expansion, which is of special importance during neuron development. By contrast, upon starvation, the intracellular decrease of malonyl-CoA and amino acids will stop the plus-end transport in order to save energy and facilitate their fusion with autophagosomes. Since malonyl-CoA levels depend on the metabolism of fatty acids and glucose, but not on amino acids (*Wolfgang and Lane, 2006*), CPT1C is unlikely to regulate protrudin-mediated LE/Lys positioning in response to changes in amino acids. Instead, amino acid mediated activation of PI3P production has been shown to play such a role (*Hong et al., 2017*).

In growing neurons, mitochondria are also targeted into actively extending axons, in order to supply ATP necessary for this high energy consuming process (*Sheng, 2017*; *Smith and Gallo, 2018*). Recently, it has been demonstrated that LE/Lys also act as mRNA translation platforms for the synthesis of new proteins necessary to sustain mitochondria function in axons (*Cioni et al., 2019*). Collectively, data from the literature indicate that LE/Lys anterograde transport is necessary for axon growth, either for rapid membrane expansion, or to provide platforms for local protein synthesis.

It is important to emphasize the relevance of malonyl-CoA as a metabolic intermediary in energy metabolism since it is at the crossroad of glucose and fatty acid metabolism. Malonyl-CoA is a derivative of glucose catabolism used for fatty acid synthesis. Its levels highly fluctuate depending on the feeding or fasting state of the individual. Specifically in the brain cortex, malonyl-CoA levels are 9 times higher after feeding than under fasting conditions (*Tokutake et al., 2010*). Malonyl-CoA levels are tightly controlled by AMPK. The involvement of the AMPK pathway in axon growth control was previously described in the literature. AICAR treatment of newly platted hippocampal neurons resulted in KIF5 phosphorylation and the prevention of PI3K enrichment at the neurite tip, which was necessary for proper axon growth (*Amato et al., 2011*). Our results identify another downstream pathway of AMPK activation, which is ACC phosphorylation and the consequent depletion of malonyl-CoA levels.

It is important to note that AMPK is activated by metabolic stresses (fasting, exercise), but also by some natural products (resveratrol), drugs (metformin), or hormones (ghrelin, adiponectin) (*Hardie, 2015*; *López, 2018*). Moreover, in neurons AMPK is also phosphorylated and activated by the calcium sensor CaMKK in response to membrane depolarization. Therefore, it will be interesting to study in the future whether all these other effectors could regulate LE/Lys transport and axon growth through the malonyl-CoA/CPT1C axis.

Another interesting finding of this work, is the interaction between CPT1C and protrudin. We propose that both proteins form stable complexes in the ER, since glucose depletion or TOFA treatment does not separate them. CPT1C could be considered a sensor of malonyl-CoA that regulates protrudin. We speculate that this regulation is mediated through the N-regulatory domain of CPT1C, which is a peptide of 50 residues that can adopt two different conformations (Nα and Nβ) as determined by NMR spectroscopy (*Rao et al., 2011*; *Samanta et al., 2014*). This N-regulatory domain switches between the two conformations depending on malonyl-CoA levels. This proposed

mechanism shows similarities with the regulation of LE/Lys transport by ORP1L-sensing of cholesterol (*Rocha et al., 2009*; *Wijdeven et al., 2015*). Both CPT1C and ORP1L would act as nutrient sensors that regulate LE/Lys transport by changes in their own conformation.

The energetic regulation of LE/Lys plus-end transport mediated by CPT1C seems to be of particular importance for corticospinal neurons, the ones with longer axons in the body, since individuals carrying the missense CPT1C$^{R37C}$ or the nonsense CPT1C$^{Q76X}$ mutations develop HSP (*Hong et al., 2019*; *Rinaldi et al., 2015*). Arg37 is predicted to establish an electrostatic interaction with Glu3, stabilizing the Nα state of CPT1C. The loss of positive charge in Arg37Cys mutation increases the distance between the two residues and probably avoids the formation of the electrostatic interaction (*Figure 10—figure supplement 1*). This would favour the extended conformation (Nβ state), leaving CPT1C$^{R37C}$ in the malonyl-CoA-unbound state, which would explain why R37C mutation results in reduced axon growth.

In summary, the present work demonstrates that CPT1C is a new regulator of LE/Lys anterograde transport by sensing malonyl-CoA levels. This is the first time that an energetic intermediate, such as malonyl-CoA, has been implicated in the regulation of LE/Lys transport. We find that protrudin is an important effector of CPT1C; although we cannot exclude that other mechanisms could be involved. Under normal nutrient conditions, CPT1C would enhance protrudin-mediated transfer of kinesin-1 to FYCO1 at the LE/Lys membrane, favoring plus-end transport. By contrast, when glucose is not available to cells or malonyl-CoA levels decrease, CPT1C would change its N-terminal conformation and slowdown the plus-end transport of LE/Lys preventing them to reach the axon tip and temporarily stop axon growth. This regulation seems of special importance in neurons with long axons, such as corticospinal motor neurons, since individuals with mutated CPT1C develop hereditary spastic paraplegia.

# Materials and methods

**Key resources table**

| Reagent type (species) or resource | Designation | Source or reference | Identifiers | Additional information |
|---|---|---|---|---|
| Cell line (*Homo sapiens*) | HeLa | Institute Curie, Paris, France | | |
| Cell line (*Homo sapiens*) | HEK293T | ATCC | CRL-1168 | |
| Strain, strain background (*Mus musculus*) | WT | Jackson Laboratory | C57BL/6J | Provided by UAB animal facility |
| Strain, strain background (*Mus musculus*) | *Cpt1c* KO | DOI: 10.1074/jbc.M111.337493 | B6129SvEv/C57BL /6J *Cpt1c$^{-/-}$* | Exons deleted: 12–15. Generated at UAB-CBATEG. 8 times backcrossed to C57BL/6J |
| Transfected construct (*M. musculus*) | EV | Addgene | #12254; pWPI-IRES-GFP | Empty vector. Lentiviral construct |
| Transfected construct (*M. musculus*) | CPT1C | DOI: 10.1016/j.molmet.2018.10.010 | pWPI-CPT1C-IRES-GFP | |
| Transfected construct (*M. musculus*) | CPT1C$^{M589S}$ | DOI: 10.1016/j.molmet.2018.10.010 | pWPI-CPT1C$^{M589S}$-IRES-GFP | |
| Transfected construct (*M. musculus*) | CPT1C$^{R37C}$ | This paper | pWPI-CPT1C$^{R37C}$-IRES-GFP; | Lentiviral construct to express mutated CPT1C$^{R37C}$ (c.109C > T). |
| Transfected construct (*M. musculus*) | shProtrudin | Tebu-Bio | Psi-LVRU6MP-shProtrudin | Cloned sequence: 5′-agaatgaggtgctgcgcag-3′ |

*Continued on next page*

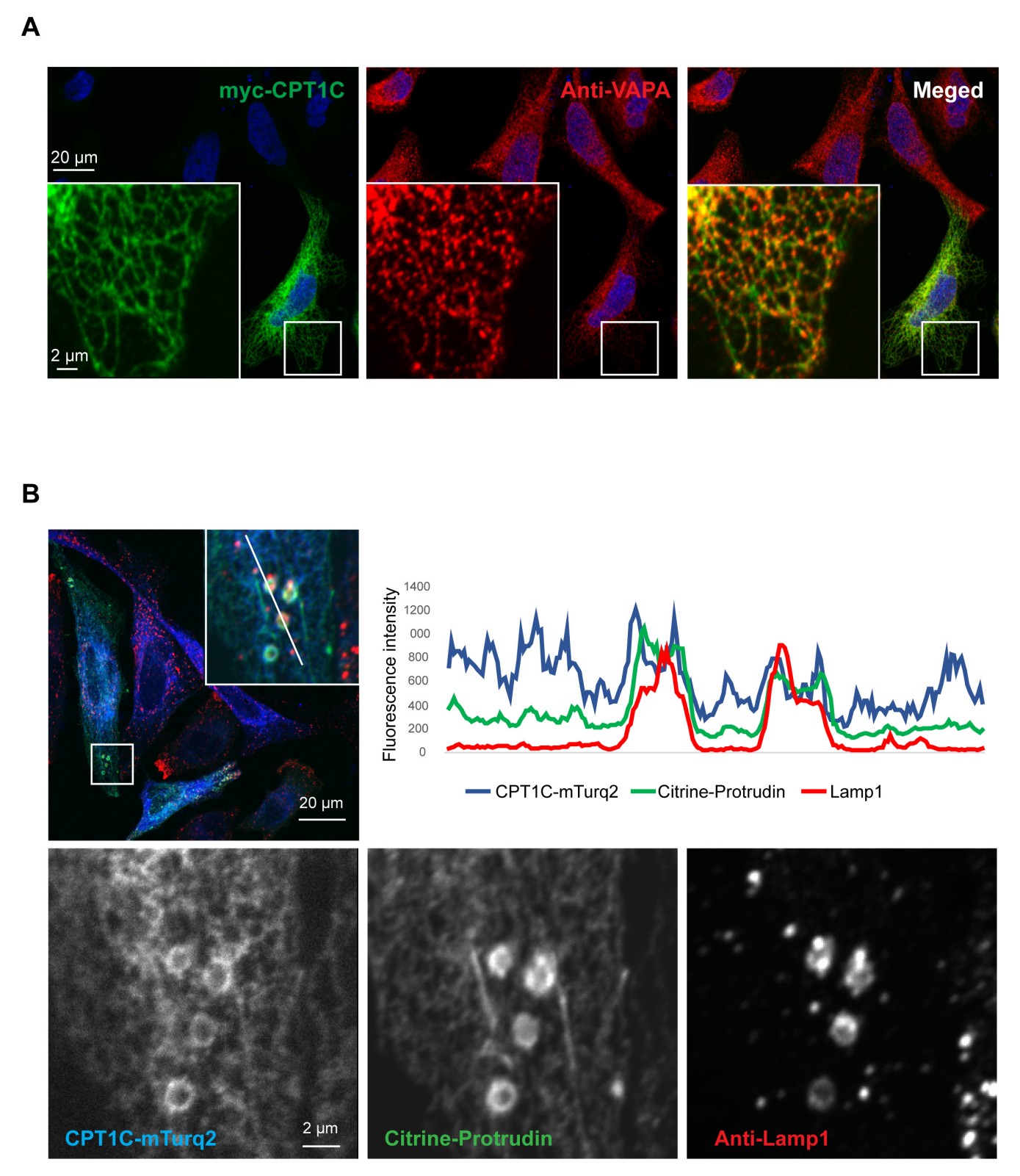

**Figure 6.** CPT1C is an ER protein localizing to ER-LE/Ly contact sites. HeLa cells were transfected with the plasmids indicated, fixed in 3% FA, stained with the indicated antibodies and analyzed by confocal microscopy. (**A**) Myc-CPT1C colocalizes with the ER resident protein VAPA. (**B**) CPT1C-mTurq2 colocalizes with the ER protein protrudin in ER-LE/Ly contact sites, but is not enriched in contact sites, contrary to protrudin (intensity line plot).

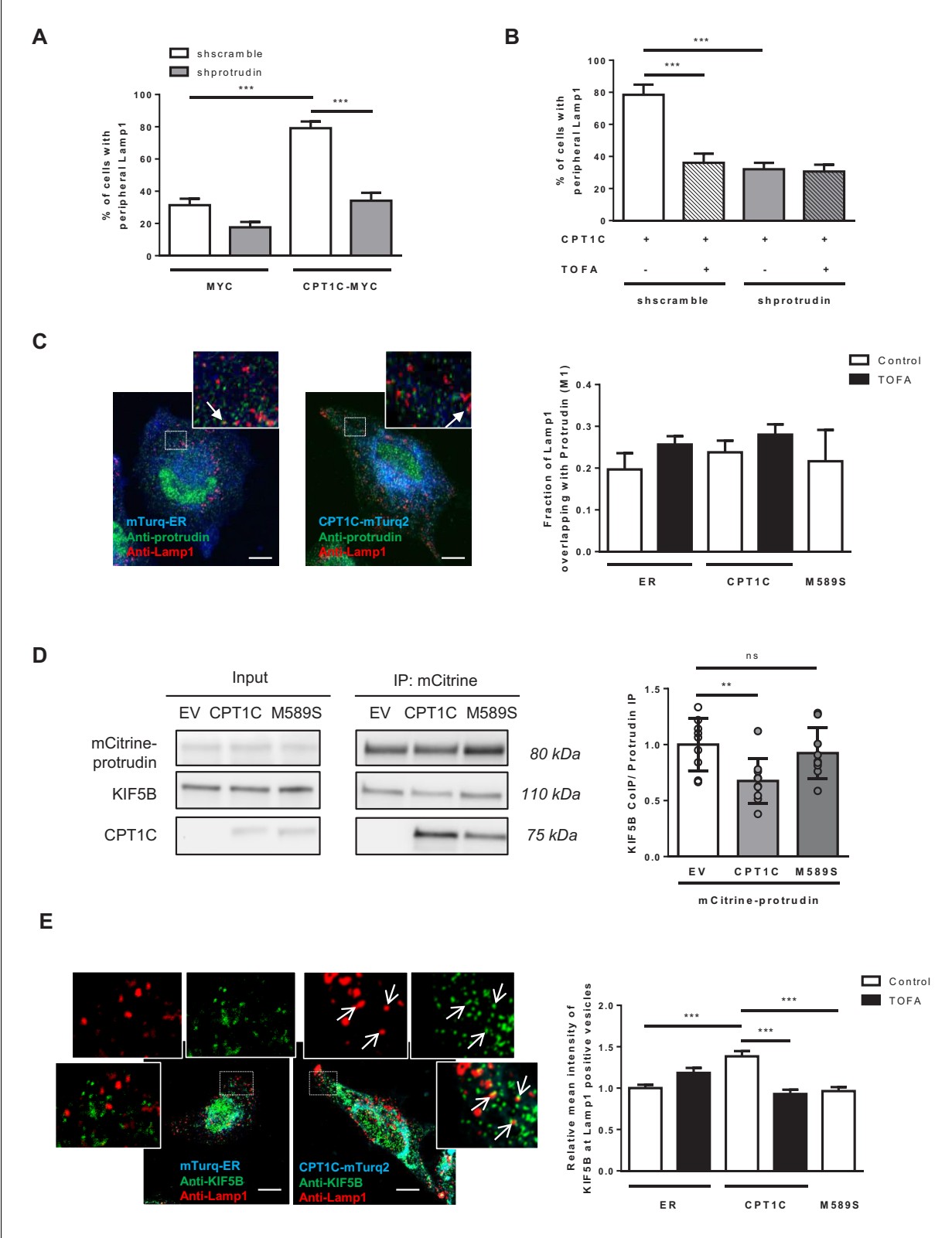

**Figure 7.** CPT1C regulates the protrudin-mediated transfer of kinesin-1 to LE/Lys in a malonyl-CoA-dependent manner. (**A**) Plasmids for protrudin silencing and CPT1C overexpression were transfected in HeLa cells at the same time. A scrambled sequence was used as a negative control of protrudin silencing, and an empty vector (Myc) was used as negative control of CPT1C overexpression. 72 hr later, Lamp1 localization was analysed by immunostaining. Results are shown as the percentage of cells with peripheral Lamp1 and given as mean ± SEM of 3 independent experiments

*Figure 7 continued on next page*

*Figure 7 continued*

performed in biological duplicates (n = 100 cells per condition were analyzed; Two-way ANOVA followed by Bonferroni's comparison test; ***p<0.001). (**B**) Percentage of cells with peripheral Lamp1 under TOFA treatment. Cells processed as in A were treated with TOFA (1 hr at 20 µg/mL) or vehicle (control, DMSO 1:500). Values are the mean ± SEM of 2 independent experiments performed in biological duplicates (n = 100 cells per condition were analyzed; Two-way ANOVA followed by Bonferroni's comparison test; ***p<0.001). (**C**) Fraction of Lamp1 overlapping with protrudin. Cells were transfected with mTurquoise-ER, CPT1C-mTurquoise2 or CPT1C$^{M589S}$ -mTurquoise2 and 48 hr later treated with TOFA for 1 hr. Then, a double immunostaining was achieved to detect protrudin and Lamp1 by confocal microscopy. Representative images (left) and quantification results (right) are shown. Results are given as Manders overlap coefficient $M_1$ (the fraction of Lamp1 in compartments containing protrudin). Values are the mean ± SEM of 3 independent experiments performed in biological duplicates (n = 30 cells per condition; One-way ANOVA followed by Bonferroni's comparison test; p>0.05). (**D**) KIF5B recruitment by protrudin. EV, CPT1C or M589S stably expressing cells were transfected with mCitrine-protrudin. 24 hr later, mCitrine-protrudin was immunoprecipitated. The expression of protrudin, KIF5B and CPT1C was evaluated by Western blot in whole lysates (input) and immunoprecipitated samples (IP). Results are given as the mean ± SD of 3 independent experiments performed in biological duplicates, triplicates or quadruplicates (n = 9–10 samples per condition; One-way ANOVA followed by Bonferroni's multiple comparison test; EV vs CPT1C: **p<0.01, and EV vs M589S: p>0.05). (**E**) Relative mean intensity of KIF5B at Lamp1-positive vesicles. Cells transfected and treated as in C were immunostained with anti-Lamp1 (in red) and anti-KIF5B (in green). Insets show magnification of a part of the region analyzed. Values are the mean ± SEM of 4 independent experiments performed in biological duplicates (n = 40 cells per condition; One-way ANOVA followed by Bonferroni's comparison test; ***p<0.001). White arrows in C and E show colocalization dots. Scale bar, 10 µm.

The online version of this article includes the following source data and figure supplement(s) for figure 7:

**Source data 1.** Source data for co-immunoprecipitation analysis (panel D of the figure).
**Figure supplement 1.** CPT1C effects on protrudin-Rab7 colocalization and FYCO1-recruitment of kinesin-1.
**Figure supplement 1—source data 1.** Source data for protrudin expression levels (panel A of the figure).

*Continued*

| Reagent type (species) or resource | Designation | Source or reference | Identifiers | Additional information |
|---|---|---|---|---|
| Transfected construct (*H. sapiens*) | EGFP-FYCO1 | DOI: 10.1038/nature14359 | pDEST-EGFP-FYCO1 (Amp$^R$) | |
| Transfected construct (*H. sapiens*) | CPT1C-mTurquoise2 | other | Empty vector: #54843 | Vector from Addgene. Cloning at D Serra's lab |
| Transfected construct (*H. sapiens*) | mTurquoise-calnexin | Addgene | #55539 | |
| Transfected construct | mTurquoise-ER | Addgene | #55550 | |
| Transfected construct | mCitrine-ER-5 | Addgene | #56557 | |
| Transfected construct (*H. sapiens*) | mCherry-FYCO1 | other | pDestm Cherry-C1-FYCO1 | From T Johansen's lab |
| Transfected plasmid (*H. sapiens*) | mCitrine-Protrudin | DOI: 10.1074/jbc.M112.419127 | | From WD Heo's lab |
| Transfected plasmid (*R. norvegicus*) | Myc-CPT1C | This paper | pCDNA3.1-Myc-His-CPT1C | Empty vector from ThermoFisher:V80020 |
| Antibody | Anti LAMP1 (rabbit polyclonal) | Sigma | L1418 | IF: (1:250) |
| Antibody | Anti Myc (mouse monoclonal) | ATCC | 9E10 | IF: (1:50) West: (1:80) |
| Antibody | Anti KIF5B (rabbit monoclonal) | Abcam | ERP10277B; ab167429 | IF: (1:100) West (1:500) |
| Antibody | Anti GFP (rabbit monoclonal) | Cell Signaling | #2956 | West: (1:500) |

*Continued on next page*

*Continued*

| Reagent type (species) or resource | Designation | Source or reference | Identifiers | Additional information |
|---|---|---|---|---|
| Antibody | Anti protrudin (rabbit polyclonal) | PTG | 12680–1-AP | IF: (1:300) |
| Antibody | Anti Rab7 (rabbit monoclonal) | Cell Signaling | D95F2;9367 | IF: (1:60) |
| Antibody | Anti CPT1C (rabbit polyclonal) | DOI: 10.1074/jbc.M707965200 | | West: (1:2000) |
| Antibody | Anti pACC (rabbit polyclonal) | Cell Signaling | #3661 | West: (1:1000) Against Ser79 |
| Antibody | Anti ACC (rabbit monoclonal) | Cell Signaling | #3676 | West: (1:1000) |
| Antibody | Anti SMI312 (mouse monoclonal) | Convance | # SMI312R | IF: (1:1000) |
| Software | Imaris | Bitplane | BPA-IM91-CL | Imaris 9.1 for Cell Biologists |

## Cell culture

Primary cortical mouse neurons were prepared from E16 WT or *Cpt1c* KO embryos, cultured and maintained as described previous (*Fadó et al., 2015*) in Neurobasal (21103049; Gibco) supplemented with B27 (17504044; Gibco), glutaMAX (35050061; Gibco) and antibiotics. For glucose starvation, DMEM without glucose (A14430-01; Gibco) was used. The cell-permeable adenosine analogue AICAR (A9978; Sigma-Aldrich) was used to selectively activate AMPK. To inhibit malonyl-CoA synthesis, 20 µg/mL 5-(tetradecyloxy)−2-furoic acid (TOFA) (ab141578; Abcam) was added. Cell lines were grown according to American Type Culture Collection. HeLa and HEK293 cells were maintained in DMEM (D5671; Sigma-Aldrich) supplemented with 10% FBS (F7524; Sigma-Aldrich), 2 mM glutamine (X0551-100; Biowest), 100 U/mL penicillin and 100 µg/mL streptomycin (P0781; Sigma-Aldrich) at 37˚C with 5% $CO_2$. Cell lines were authenticated by genotyping and regularly tested for mycoplasma contamination. All the experiments were performed with cells at 4–14 passages. Malonyl-CoA was purchased from Sigma-Aldrich (M4263).

## Lentiviral particles

To overexpress CPT1C, CPT1C$^{M589S}$ or CPT1C$^{R37C}$ in primary culture of neurons, plasmids pWPI-IRES-GFP (empty vector EV), pWPI-CPT1C-IRES-GFP, pWPI-CPT1CM589S-IRES-GFP (*Rodríguez-Rodríguez et al., 2019*), and pWPI-CPT1CR37C-IRES-GFP (generated by site-directed mutagenesis) were used to generate lentiviral particles, as previously described (*Fadó et al., 2015*). Cells were transduced with low virus titles (multiplicity of infection, m.o.i. = 2) for the infection of cultured cortical neurons.

## Plasmids and transfection

Human CPT1C-mTurquoise2 and CPT1C-SYFP2 were a gift from D Serra´s group (University of Barcelona, Spain). Human mCherry-FYCO1 was a gift from T Johansen´s group (University of Trømso, Norway). mCitrine-protrudin was a gift from W Do Heo (*Gil et al., 2012*). mTurquoise-calnexin, mTurquoise-ER, mCitrine-ER-5 were from Addgene (plasmids #55539,#55550 and #56557, respectively). Human atlastin-1-mTurquoise2, protrudin-SYFP2, GluA1-SYFP2, Myc-CPT1C were generated by standard molecular biological methods. The ORFs of atlastin-1, protrudin and GluA1 were obtained by RT-PCR from the human neuroblastoma cell line SH-S5Y5 (Sigma (ECACC) reference 94030304-1VL), using TRIzol (15596018; Life Technologies) for RNA isolation, Mu-MLV (#3022–1 Lucigen) as a reverse transcriptase, and the Q5 DNA polymerase (ref. M0491L, New England Biolabs). Plasmids were verified by sequencing and restriction digestion. CPT1C$^{M589S}$-mTurquoise2 was generated by site-directed mutagenesis. GFP-protrudin was generated as described in *Raiborg et al. (2015)*. FuGENE6 (E2692, Promega) was used for transient transfection in HeLa cells, and the calcium phosphate method for HEK293 cells. For protrudin silencing, the sequence 5'-

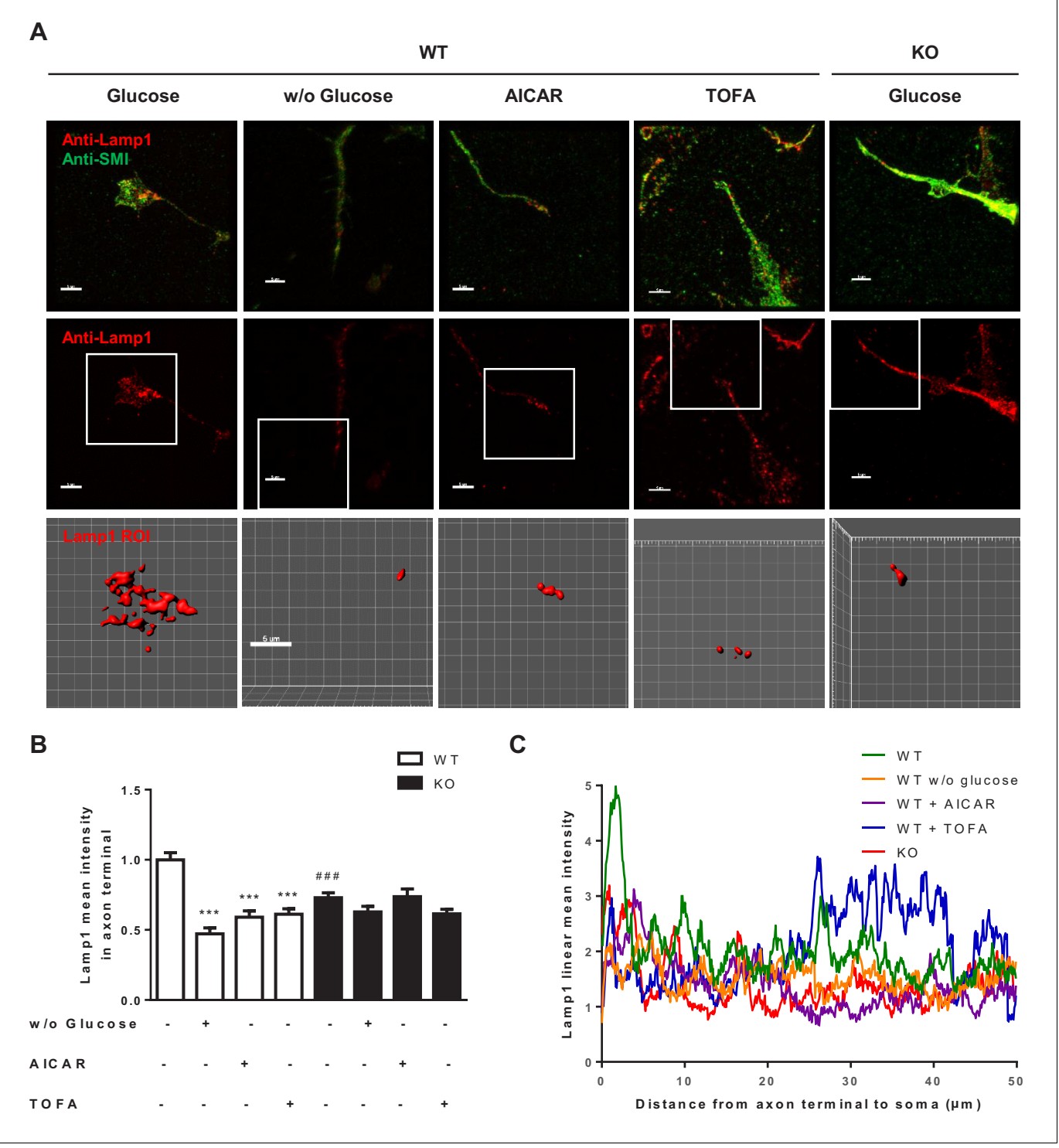

**Figure 8.** Metabolic stress reduces Lamp1 in axon terminals through CPT1C. 3DIV primary cortical neurons derived from WT and *Cpt1c* KO embryos were treated as in *Figure 2*. 24 hr later (at 4DIV), cells were fixed and processed by immunostaining with anti-Lamp1 and anti-SMI-321 (axon marker). (A) Representative images of the Lamp1 staining at the axon terminal. The region of interest (ROI) selected for the analysis is shown in the amplified square. (B) Lamp1 mean intensity at the axon terminal. The same threshold was used in all the samples. Results are shown as the mean ± SEM of 3 independent experiment performed in biological duplicates (n = 35–60 cells per condition; Two-way ANOVA followed by Bonferroni's multiple comparison test; ***$p<0.001$ *versus* control treatment and ###$p<0.001$ *versus* WT within the same treatment). (C) Lamp1 linear intensity profile. The sum intensity histograms from Z-projection images were analyzed. Due to different axon lengths between groups, only the last 50 μm from the axon

*Figure 8 continued on next page*

*Figure 8 continued*

terminal to the soma was measured. The graph shows the values of the mean fluorescence intensity measured every 0.1 μm from one representative experiment out of 14–19 cells per condition.

The online version of this article includes the following figure supplement(s) for figure 8:

**Figure supplement 1.** Metabolic stress decreases lamp1 in axon terminals through CPT1C.

---

agaatgaggtgctgcgcag-3' (*Hong et al., 2017*) was cloned in the vector psi-LVRU6MP (Tebu-Bio) to generate the plasmid shprotrudin. As a silencing control (shscramble), the random sequence 5'-gcttcgcgccgtagtctta-3' (CSHCTR001-LVRU6MP, Tebu-Bio) was used.

## Generation of stable cell lines

HeLa stably expressing mouse CPT1C or CPT1C$^{M589S}$ were generated by lentiviral transduction of pWPI-CPT1C-IRES-GFP (called CPT1C cells) and pWPI-CPT1CM589S-IRES-GFP (M589S cells), respectively. Cells were transduced with low virus titles (multiplicity of infection, m.o.i. = 5) and stable cells showing green fluorescence isolated by cell-sorter (Facs Aria Fusion, Becton Dickinson, San José, California, EU). GFP was excited at 488 nm and green fluorescence collected at 530/30 nm. pWPI-IRES-GFP empty vector was used to obtain control cells (EV cells). No more than 8 passages were made after its generation.

## Immunostaining

Immunostaining was performed as described in previously (*Raiborg et al., 2015*). Mouse anti-neurofilament SMI-312R was from Covance. Rabbit anti-LAMP1 (L1418) was from Sigma. Mouse anti-myc (9E10) was from ATCC. Rabbit anti KIF5B (EPR10277B), chicken anti-GFP (ab13970) and rabbit anti-Myc (ab9106) were from Abcam. Rabbit anti-protrudin (12680–1-AP) was from PTG. Rabbit anti-Rab7 (D95F2;9367) were from Cell Signaling Technologies. Secondary antibodies used for confocal studies were obtained from Jackson ImmunoResearch Laboratories and Thermo Fisher.

## Immunoblotting

Western blot was performed as previously described (*Rodríguez-Rodríguez et al., 2019*). Rabbit anti-CPT1C was developed in our laboratory (*Sierra et al., 2008*). Rabbit anti-ACC (3676), rabbit anti-pACC (Ser79) (3661) and rabbit anti-GFP (2956) were from Cell Signaling. Rabbit anti-protrudin (12680–1-AP) was from PTG. Mouse anti-GAPDH (ab36845) was from Abcam. Mouse anti-actin (ma1-91399) was from Fisher Scientific. Horseradish peroxidase-conjugated secondary antibodies, anti-mouse or anti-rabbit were from DAKO (Glostrup, Denmark). GAPDH was used as an endogenous control to normalize protein expression levels. In all the figures showing images of gels, all the bands for each picture come from the same gel.

## Immunoprecipitation assay

Transfected HeLa cells were collected and GFP- or citrine-containing proteins were immunoprecipitated from the supernatant of cellular lysates using GFP-Trap assay (it possesses high specificity also for citrine) following manufacturer's protocol (GTA-100; Chromotek). However, one extra centrifugation of 100000 xg was performed to eliminate non solubilized proteins from the supernatant of the cellular lysates, before the immunoprecipitation. Finally, immunoprecipitated proteins and their interactors were detected by Western blot in whole lysates (inputs) and immunoprecipitated samples (IP).

## FRET analysis

All FRET imaging experiments were performed on a Leica TCS SP2 microscope. A 63x/1.40 N.A. objective with immersion oil and argon lasers with 458 and 514 lines (for donor, mTurquoise2, and acceptor, YFP, respectively) were used for all image acquisition. The pinhole size was set to 2 (airy units, UA) and the ROI to 5 × 5 (μm). 458 nm-laser power was set up to 23–60% and 514 nm laser up to 5–10% so that no pixel saturation occurred. The number of bleaches was between 3 and 4, all at 40%. For the calculation of FRET efficiency from acceptor to donor, we used the following formula operated at pixel basis,

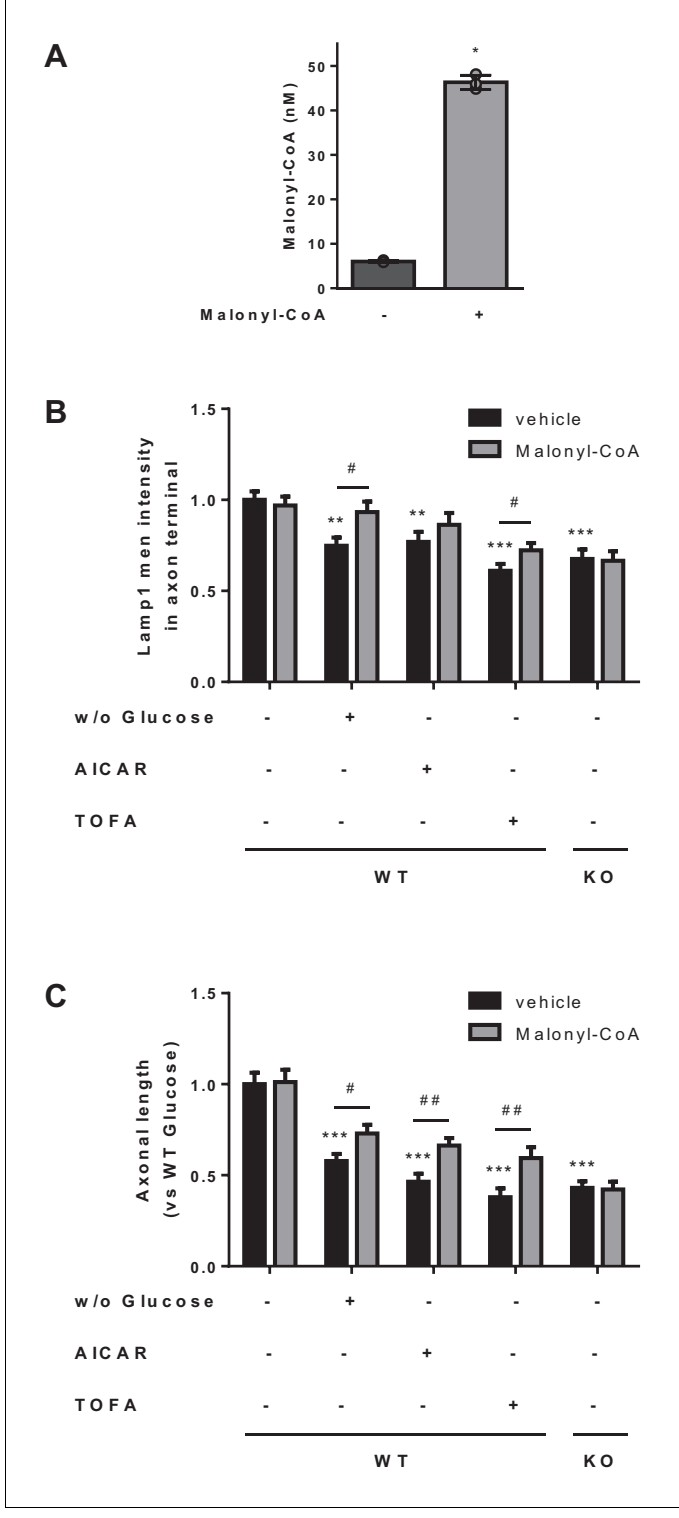

**Figure 9.** Malonyl-CoA partially rescues the effects of metabolic stress on axonal length and LE/Lys localization. (A) Intracellular malonyl-CoA levels. HeLa cells were treated with exogenous malonyl-CoA (200 µM) for 2 hr and washed 3 times with PBS, collected and processed to determine intracellular levels of malonyl-CoA. Raw data are the mean of technical duplicates. Values are the mean ± SD of triplicates (n = 3 samples per condition; Mann-Whitney U test; p=0.05). (B–C) Effect of malonyl-CoA treatment in Lamp1 intensity at the axon terminal and in axonal length. Cultured cortical neurons were treated simultaneously with malonyl-CoA (2 hr at 200 µM) and the same conditions used in *Figure 2*. Lamp1 mean intensity at axon terminal and mean axonal length are shown in B

*Figure 9 continued on next page*

*Figure 9 continued*

and C, respectively. Results are given as the mean ± SEM of 2 independent experiment performed in biological duplicates (n = 40 cells per condition). Two-way ANOVA followed by Bonferroni's comparison test was used to compare WT in control conditions *versus* each treatment (\*p<0.01 and \*\*\*p<0.001 *versus* WT in control conditions); Student's t test was applied to compare each treatment with or without malonyl-CoA (#p<0.05 *versus* the same condition without malonyl-CoA).

The online version of this article includes the following source data for figure 9:

**Source data 1.** Source data for malonyl-CoA analysis in cells (panel A of the figure).

$$E = I_{fret}/I_d = (I_a - I_{dcrossem - I_{crosses}})/I_d$$

where $I_{fret}$ refers to FRET intensity, $I_d$ refers to donor intensity in the absence of acceptor, $I_a$ is the acceptor intensity in the presence of donor, $I_{dcross\ em}$ is the fraction of donor emission cross-emitted into the acceptor channel and $I_{cross\ ex}$ is the fraction of acceptor intensity cross-excited by donor excitation.

## Confocal fluorescence microscopy

Confocal images of HeLa cells were obtained using LSM710 or LSM780 confocal microscope (Carl Zeiss) equipped with an Ar-laser multiline (458/488/514 nm), a DPSS-561 10 (561 nm), a laser diode 405–30 CW (405 nm), and an HeNe laser (633 nm). The objective used was a Plan-Apochromat 63×/ 1.40 N.A. oil DIC III (Carl Zeiss). Confocal images of cultured neurons were obtaines using Leica DMi8 confocal microscope, using a Plan-Apochromat 63×/1.40 N.A. oil objective. For quantification, sets of cells were cultured and stained simultaneously, and imaged using identical settings. Measurements were performed with the Imaris 9.2 image processing package (Bitplane). For colocalization studies, the Manders overlap coefficient was calculated using ImageJ software. Usually a region of the cytoplasm between the nucleus and the tip of a protrusion, and not crowded with vesicles, was selected as a region of interest in Manders analysis. For the analysis of percentage of cells with Lamp1 in their protrusion tips, ER, CPT1C, M589S and protrudin positive cells were picked based on intensity of the stained turquoise and citrine tags respectively. Peripheral Lamp1 clusters were detected by setting threshold values of intensity and size using Cell and Surface tools from Imaris software. For circularity analysis, NIS-Elements software was used. For the analysis of axon outgrowth in cortical neurons, several images of the same cell were taken and merged to follow the trajectory of the axons stained with the neurofilament marker SMI-312R, and the axonal lengths were measured using Tracing tool of Imaris software. For the measurement of Lamp1 intensity at the growth cone about 20 z-stacks (every 0.5 µm) per axon terminal (stained with SMI-312R) were taken and the mean intensity, volume and total intensity of the most apical Lamp1-positive dot accumulation at the axon terminal was determined using Surface tool of Imaris software. The same threshold values were used in all the samples. For the analysis of Lamp1 linear fluorescence profile, the sum intensity histogram from Z-projection images for each cell was obtained using Fiji software. 1 µm line width was drawn along the axon from the tip towards the soma and only the terminal 50 µm were analyzed (values were taken every 0.1 µm).

## Live-cell imaging

Live cell imaging to monitor vesicle movement in HeLa cells was done as previously described (*Raiborg et al., 2015*). A Delta Vision Deconvolution microscope (Applied Precision, GE Healthcare) with a x60 objective without binning was used. Images were acquired at 0.5 Hz for 2 min. Vesicles were tracked using the ImageJ plugin 'Manual tracking' and the analysis was performed with Chemotaxis (IBIDI) software.

## Analysis of malonyl-CoA levels

About 3.5 million of EV, CPT1C and M589S stably expressing HeLa cells were washed with PBS and collected by centrifugation. Then, cells were suspended in cold methanol and broken by sonication. Malonyl-$^{13}C_3$-CoA internal standard was added (final concentration 0.5 µM) and mixed by vortex. Finally, samples were centrifuged and supernatants were concentrated on ice-water bath under

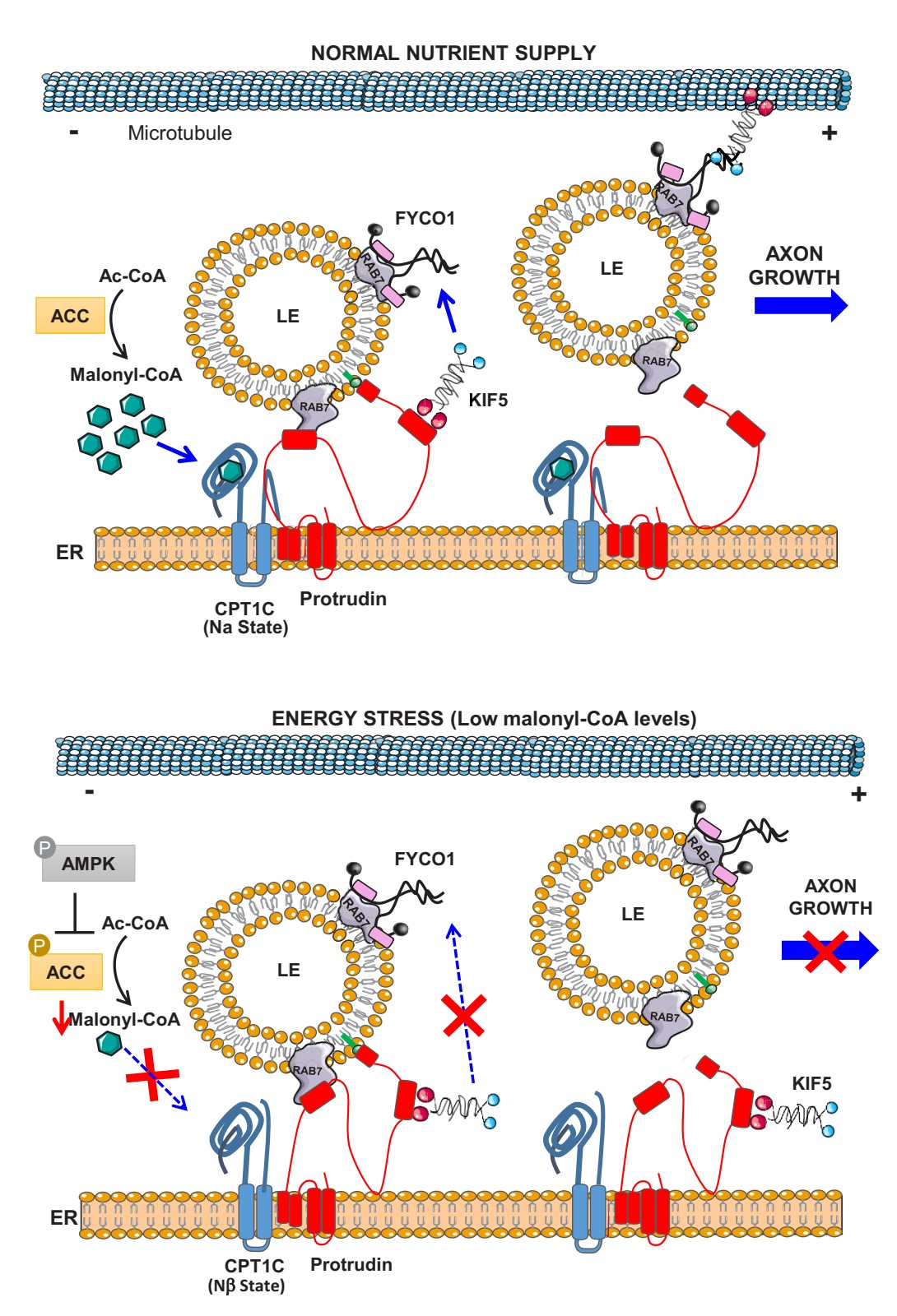

**Figure 10.** Model for the role of CPT1C in LE/Lys anterograde transport and axon growth. In normal nutrient conditions, CPT1C is bound to malonyl-CoA and its N-terminal domain acquires the folded conformation (Nα), which enhances the transfer of kinesin-1 (KIF5) from protrudin to FYCO1, which promotes the plus-end transport of LE/Lys and axon growth. However, under energy stress, such as glucose depletion or the activation of AMPK pathway, N-regulatory domain of CPT1C switches to its extended conformation (Nβ). In this state, CPT1C does not enhance the transfer of kinesin-1 to

*Figure 10 continued*

FYCO1, and in consequence, the anterograde transport of LE/Lys is not promoted and axon growth arrested. CPT1C and protrudin are bound in both situations, but the way they interact with each other changes with metabolic stress.

The online version of this article includes the following figure supplement(s) for figure 10:

**Figure supplement 1.** Effect of R37C mutation on the N-terminal domain of CPT1C.

gentle $N_2$ current followed by a final freeze-drying step. Malonyl-CoA concentration in cellular extracts was determined by LC-MS/MS method as previously described (*Schriewer et al., 2017*). The method was modified utilizing isotopic dilution analysis with $^{13}$C-labeled internal standard for quantification instead of standard addition approach.

## Statistical analysis

As indicated in each figure legend, the results were obtained from 2 to 4 independent experiments performed, at least, in biological duplicates and are given as the mean ± SEM or SD. When individual cells were analysed, we decided to evaluate between 5 and 10 per condition for FRET assays and between 30 and 100 cells for immunostaining studies, at least, seeded in two independent coverslips per experiment. Exceptionally, the outliers were excluded if the values were above or below the mean ±two standard deviations. Statistical analysis was performed using PRISM (GraphPad Software). Data normality was determined according to Shapiro-Wilk test and D'Agostino and Pearson test. Significances between groups were detected using either Student's t test or a Mann-Whitney U test (parametric and non-parametric, respectively) and a 95% confidence interval. For comparisons among 3–4 groups, ANOVA was performed (One-way or Two-way), followed by the Bonferroni's multiple comparison post-test.

## Acknowledgements

We are very grateful to Eva M Wenzel for her help in the settings of live imaging experiments and critical discussion. This work was supported by the Ministry of Spain (MINECO) (Grants SAF2014-52223-C2-1-R to DS, SAF2017-83813-C3-1-R to DS, SAF2014-52223-C2-2-R to NC, and SAF2017-82813-C3-3R to NC, cofunded by the European Regional Development Fund [ERDF]), the *Centro de Investigación Biomédica en Red Fisiopatología de la Obesidad y la Nutrición* (CIBEROBN) (Grant CB06/03/0001 to DS), the *Generalitat de Catalunya* (2014SGR465 to DS and NC), *Fundació La Marató de TV3* (Grant 87/C/2016 to DS and NC). POH thanks the Fonds der Chemischen Industrie (FCI, Frankfurt am Main, Germany) for a PhD scholarship. The authors declare no competing financial interests.

## Additional information

### Funding

| Funder | Grant reference number | Author |
| --- | --- | --- |
| Ministerio de Economía y Competitividad | SAF2014-52223-C2-2-R | Núria Casals |
| Ministerio de Economía y Competitividad | SAF2017-82813-C3-3R | Núria Casals |
| Centro de Investigación Biomédica en Red de Obesidad y Nutrición | CB06/03/0001 | Dolors Serra |
| Generalitat de Catalunya | 2014SGR465 | Dolors Serra Núria Casals |
| Fonds der Chemischen Industrie | PhD Scholarship | Patrick O Helmer |
| Fundació la Marató de TV3 | 87/C/2016 | Dolors Serra Núria Casals |

| Agència de Gestió d'Ajuts Universitaris i de Recerca | PhD fellowship | Marta Palomo-Guerrero |
| Ministerio de Economía y Competitividad | SAF2017-83813-C3-1-R to DS | Dolors Serra |
| Ministerio de Economía y Competitividad | SAF2014-52223-C2-1-R | Dolors Serra |

The funders had no role in study design, data collection and interpretation, or the decision to submit the work for publication.

## Author contributions

Marta Palomo-Guerrero, Conceptualization, Formal analysis, Validation, Investigation, Visualization; Rut Fadó, Conceptualization, Formal analysis, Supervision, Validation, Investigation, Visualization, Methodology; Maria Casas, Marta Pérez-Montero, Miguel Baena, José Luis Domínguez, Investigation, Methodology; Patrick O Helmer, Resources, Validation, Investigation, Methodology; Aina Roig, Resources, Investigation; Dolors Serra, Resources; Heiko Hayen, Validation, Investigation, Methodology; Harald Stenmark, Camilla Raiborg, Conceptualization, Resources, Supervision, Methodology; Núria Casals, Conceptualization, Resources, Supervision, Funding acquisition, Project administration

## Author ORCIDs

Maria Casas https://orcid.org/0000-0002-5246-8874
Patrick O Helmer https://orcid.org/0000-0002-1199-6291
Dolors Serra http://orcid.org/0000-0002-4936-4206
Heiko Hayen http://orcid.org/0000-0002-4074-8545
Núria Casals https://orcid.org/0000-0002-6719-4300

## Ethics

Animal experimentation: All animal procedures were performed in agreement with European guidelines (2010/63/EU) and approved by the Universitat Autonoma de Barcelona Local Ethical Committee.

## Decision letter and Author response

Decision letter https://doi.org/10.7554/eLife.51063.sa1
Author response https://doi.org/10.7554/eLife.51063.sa2

# Additional files

## Supplementary files
• Transparent reporting form

## Data availability

All data generated or analysed during this study are included in the manuscript and supporting files. Source data files have been provided for Figures 2A, 3C, 3D, 3E, 3F, 4A, 7D, 9A and Figure 7—figure supplement 1A.

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
