## [Decision Letter]

**Acceptance summary:**

This study uncovers an unexpected intersection between axon outgrowth and fatty acid metabolism in which malonyl-CoA and its sensing by CPT1C control lysosome motility. The data are of high interest to cell and neurobiologists alike and of relevance for biomedicine as CPT1C is mutated in a subset of human hereditary spastic paraplegia (HSP) patients. The story is presented in a compelling manner. The data, of which there are many pieces, support an intriguing and complex signaling path. Although the precise mechanism by which malonyl CoA alters the interaction between CPT1C and protrudin remains to be worked out, the delineation of the pathway is a significant advance.

**Decision letter after peer review:**

Thank you for submitting your article "Sensing of nutrients by CPT1C regulates late endosome/lysosome anterograde transport and axon growth" for consideration by *eLife*. Your article has been reviewed by three peer reviewers, one of whom is a member of our Board of Reviewing Editors, and the evaluation has been overseen by Suzanne Pfeffer as the Senior Editor. The following individual involved in review of your submission has agreed to reveal their identity: Volker Haucke (Reviewer #2).

The reviewers have discussed the reviews with one another and the Reviewing Editor has drafted this decision to help you prepare a revised submission.

This is an interesting paper that uncovers an unexpected intersection between axon outgrowth and fatty acid metabolism in which malonyl-CoA and its sensing by CPT1C control lysosome motility. The data are of high interest to cell and neurobiologists alike and of relevance for biomedicine as CPT1C is mutated in a subset of human hereditary spastic paraplegia (HSP) patients. The story is presented in a compelling manner. The data presented, of which there are many pieces, are largely consistent with the proposed model. Although the precise mechanism by which malonyl CoA alters the interaction between CPT1C and protrudin remains to be worked out, the take home, as is, is a significant advance warranting publication in *eLife* if the following issues can be adequately addressed. Reviewers identified two major issues that need to be resolved in a resubmitted manuscript prior to a final decision.

Essential revisions:

1) A key result, demonstrating physical interaction between CPT1C and protrudin (Figure 3), is not convincing. In Figure 3A, the CPT1C signal in the citrine-protrudin IP is very faint and also in the midst of a high background of the IP antibody in the sample. One worries that the weak signal in the blot is just incompletely reduced antibody from the IP. In contrast, in Figure 3F, the CPT1C signal looks much stronger and much cleaner (were different antibodies used?), but there are no controls shown. In light of this, Figure 3A should be repeated, preferably using an antibody for the blot that does not pick up the antibody used for the IP. Additionally, a negative control using a mCitrine-tagged ER membrane protein must be used to ensure that this IP-blot result is not simply due to incomplete solubilization. (Ideally, this would be a mutated version of protrudin that lacks the CPT1C binding site. Such a construct could then be used in the rescue experiments thereby validating not just the physical interaction but also its functional importance. This is mentioned as encouragement because it would greatly strengthen the study at the mechanistic level. However, because the binding site will need to be mapped, it is not a requirement for resubmission.)

2) The subcellular localization of CPT1C remains unclear. In Supplementary Figure 2B, C it is shown that CPT1C fails to colocalize with LAMP1, while its binding partner protrudin partially overlaps with LAMP1. Overexpressed CPT1C in the images shown seems rather diffusive, while the biochemical data suggest that it may be located at the ER. It is suggested that the localization of CPT1C be studied in more detail at near endogenous expression levels, either by endogenous antibodies or low-level lentiviral expression. Is CPT1C detectable at ER-lysosome membrane contact sites?

---

## [Author Response]

Essential revisions:1) A key result, demonstrating physical interaction between CPT1C and protrudin (Figure 3), is not convincing. In Figure 3A, the CPT1C signal in the citrine-protrudin IP is very faint and also in the midst of a high background of the IP antibody in the sample. One worries that the weak signal in the blot is just incompletely reduced antibody from the IP. In contrast, in Figure 3F, the CPT1C signal looks much stronger and much cleaner (were different antibodies used?), but there are no controls shown. In light of this, Figure 3A should be repeated, preferably using an antibody for the blot that does not pick up the antibody used for the IP. Additionally, a negative control using a mCitrine-tagged ER membrane protein must be used to ensure that this IP-blot result is not simply due to incomplete solubilization. (Ideally, this would be a mutated version of protrudin that lacks the CPT1C binding site. Such a construct could then be used in the rescue experiments thereby validating not just the physical interaction but also its functional importance. This is mentioned as encouragement because it would greatly strengthen the study at the mechanistic level. However, because the binding site will need to be mapped, it is not a requirement for resubmission.)

As suggested by the reviewers, we have performed a new IP using mCitrine-ER-5 (Addgene) as a control. This plasmid encodes citrine with the tag KDEL, which drives the protein to the ER membrane. Hela cells were co-transfected with mCitrine-ER-5 or mCitrine-Protrudin and Myc-CPTC, and were immunoprecipitated by the GFP-trap assay. A faint signal corresponding to CPT1C was observed in the IP fraction of mCitrine-KDEL, probably due to incomplete solubilization or nonspecific binding. However, a far more intense band of CPT1C was observed in the mCitrine-protrudin IP. This experiment has been performed in triplicates, with similar results. The old Figure 3A has been replaced by the new Figure 3A.

We find the recommendation given by the reviewers of mapping the region of protrudin involved in the binding to CPT1C very interesting. However, we have not conducted this study because it exceeds the time given for the revision.

2) The subcellular localization of CPT1C remains unclear. In Supplementary Figure 2B, C it is shown that CPT1C fails to colocalize with LAMP1, while its binding partner protrudin partially overlaps with LAMP1. Overexpressed CPT1C in the images shown seems rather diffusive, while the biochemical data suggest that it may be located at the ER. It is suggested that the localization of CPT1C be studied in more detail at near endogenous expression levels, either by endogenous antibodies or low-level lentiviral expression. Is CPT1C detectable at ER-lysosome membrane contact sites?

We agree with the editors and reviewers that the subcellular localization of CPT1C remained unclear and that images from Supplementary Figure 2B were not conclusive. To resolve this issue, we have repeated the experiments in cells with low CPT1C-mTurq and mCitrine-protrudin overexpression levels, and posterior staining of Lys with anti-Lamp1 (new Figure 6). Moreover, we have confirmed CPT1C localization in the ER, using VAPA as a marker. Our new data demonstrate that CPT1C is nicely localized in the ER, as expected. Moreover, CPT1C is found in ER-lysosome contact sites, even though it is not enriched there, compared to protrudin, which is highly enriched in the contact sites. This agrees with the fact that CPT1C does not form the contact sites, and only regulates their activity. In light of these findings, we have decided to remove the older Supplementary Figure 2 and substitute it by the new Figure 6.